# Accelerating glacier volume loss on Juneau Icefield driven by hypsometry and melt-accelerating feedbacks

Bethan Davies [1] ✉, Robert McNabb [2], Jacob Bendle[3], Jonathan Carrivick [4], Jeremy Ely [5], Tom Holt[6], Bradley Markle[7], Christopher McNeil [8], Lindsey Nicholson[9] & Mauri Pelto[10]

Globally, glaciers and icefields contribute significantly to sea level rise. Here we show that ice loss from Juneau Icefield, a plateau icefield in Alaska, accelerated after 2005 AD. Rates of area shrinkage were 5 times faster from 2015–2019 than from 1979–1990. Glacier volume loss remained fairly consistent (0.65–1.01 km$^3$ a$^{-1}$) from 1770–1979 AD, rising to 3.08–3.72 km$^3$ a$^{-1}$ from 1979–2010, and then doubling after 2010 AD, reaching 5.91 ± 0.80 km$^3$ a$^{-1}$ (2010–2020). Thinning has become pervasive across the icefield plateau since 2005, accompanied by glacier recession and fragmentation. Rising equilibrium line altitudes and increasing ablation across the plateau has driven a series of hypsometrically controlled melt-accelerating feedbacks and resulted in the observed acceleration in mass loss. As glacier thinning on the plateau continues, a mass balance-elevation feedback is likely to inhibit future glacier regrowth, potentially pushing glaciers beyond a dynamic tipping point.

Globally, mountain glaciers and ice caps dominate loss of land ice[1], and were responsible for 21% of observed global sea level rise from 1993–2017[1]. The largest contributions (8 mm from 1961–2016) were from Alaskan and Western Canadian glaciers[1,2], which hold a large volume of ice (46.4 ± 15.3 mm sea-level equivalent)[3], and which will contribute to sea level rise for centuries to come[4,5]. Alaska will likely remain the largest regional contributor to 2100 AD[6]. Model projections indicate that, with emissions policies defined under the Paris Agreement, land ice will drive 25 cm (11, 40 cm at 5th, 95th percentiles) sea-level rise by 2100 AD. Of this, 13 cm will come from glaciers, 25% of which will come from Alaska alone[4]. Under the higher-emission scenario RCP 4.5, around 30% of glacier ice will have disappeared from Alaska by the year 2100 (relative to 2015 AD), causing 17 ± 4 mm of global sea level rise[7].

Alaskan icefields may be particularly vulnerable to accelerated melt as the climate warms. Firstly, a large ice volume in Alaska is in hypsometrically top-heavy or plateau icefields (cf.[3,8], Fig. 1) with a low-slope accumulation area, making them likely vulnerable to small changes in equilibrium line altitude (ELA)[8,9]. Secondly, a decrease in height of a flatter icefield or ice cap will theoretically result in a decrease in surface mass balance (SMB) as the ice surface lowers into warmer air, which reinforces itself as an SMB-elevation positive feedback[10]. Thirdly, due to an absence of higher topography, flatter ice caps and icefields cannot retreat to higher elevations and find a new equilibrium[11]. Icefields therefore are likely to exhibit threshold behaviour[10,12], with tipping points for rapid and irreversible recession. However, a paucity of long, multi-decadal empirical observations of icefield change impedes our ability to accurately constrain and

[1]School of Geography, Politics and Sociology, Newcastle University, Newcastle-upon-Tyne, UK. [2]School of Geography and Environmental Sciences, Ulster University, Colraine, Northern Ireland, UK. [3]Geological Survey of Norway, Trondheim, Norway. [4]School of Geography and water@leeds, University of Leeds, Leeds, UK. [5]Department of Geography, University of Sheffield, Sheffield, UK. [6]Centre for Glaciology, Aberystwyth University, Aberystwyth, UK. [7]Geological Sciences, University of Colorado Boulder, Boulder, CO, USA. [8]U.S. Geological Survey, Alaska Science Center, Anchorage, AK, USA. [9]Department of Atmospheric and Cryospheric Sciences, Universität Innsbruck, Innsbruck, Austria. [10]Nichols College, Dudley, Massachussets, USA. ✉e-mail: bethan.davies@newcastle.ac.uk

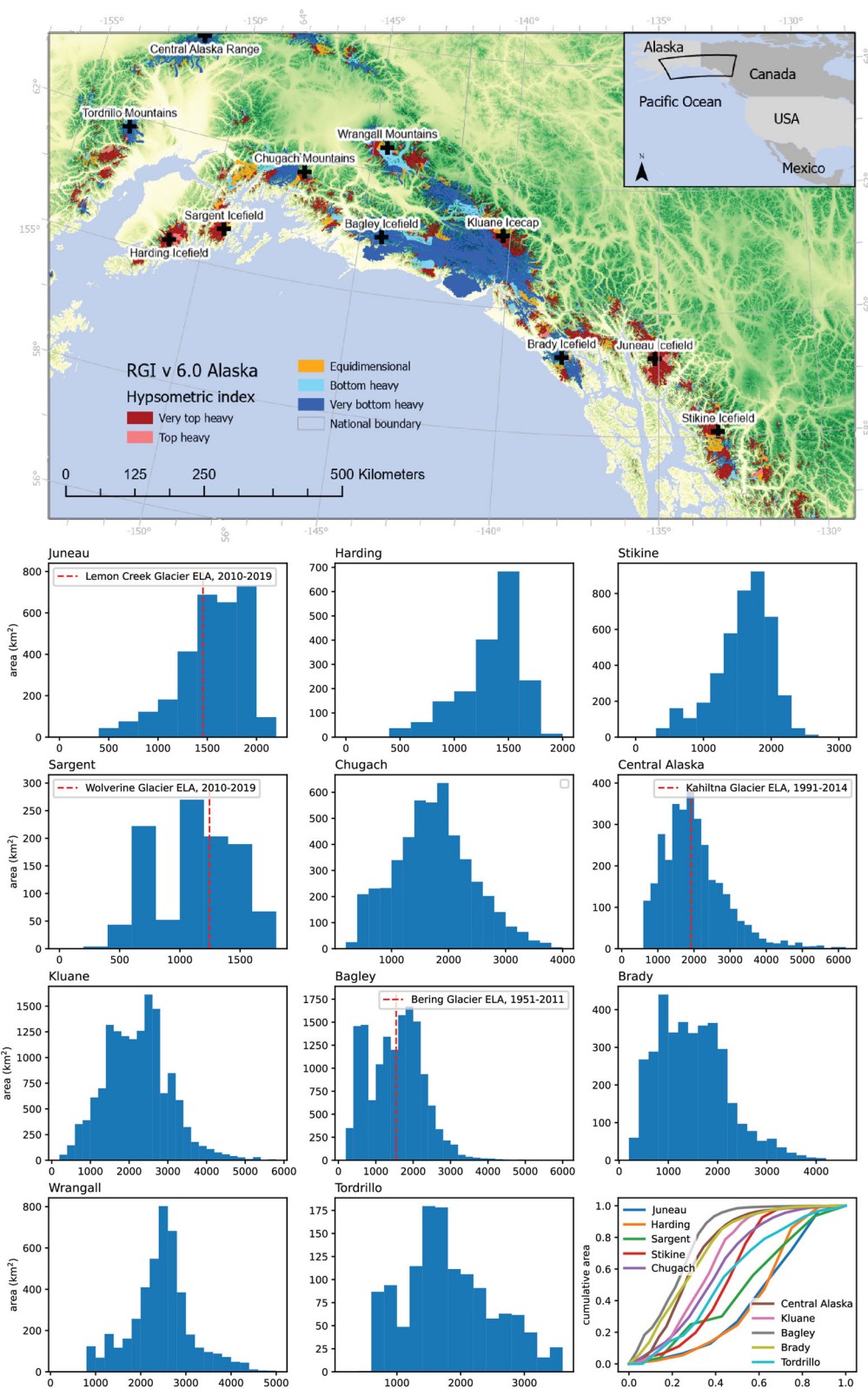

**Fig. 1 | Hypsometry of major Alaskan icefields.** Top: hypsometric index of individual glaciers in the RGI (Randolph Glacier Inventory) v6.0[67]. Hypsometric index defined after ref. 63. Visualised on background of void-filled 3″ Viewfinder Panorama DEMs (Digital Elevation Models), comprising SRTM DEM (Shuttle Radar Topography Mission Digital Elevation Model) south of 60°20'N. Below: area-elevation histograms for major ice bodies in Alaska. Measured Equilibrium Line Altitudes (ELAs)[25,33,35,94,95] shown with red dashed line. Juneau, Kluane, Brady, Sargent and Bagley icefields all have a significant low-slope plateau in their accumulation areas. Juneau, Stikine, Harding, Sargent and parts of Kluane Icefield are all very top-heavy icefields. Bottom right: normalised cumulative area-elevation plots for each icefield. The top-heavy Juneau, Stikine, Harding and Sargent icefields alone make up 2589 km³ of ice.

evidence these processes and identify tipping points and thresholds of change. Understanding the physical processes that affect the different way icefields and ice caps respond to climate change is key to improving our ability to estimate icefield response to future climate change[13–15].

Upskilling our ability to predict future glacier behaviour using model-data comparisons requires detailed observations for model parameterisation and calibration or weighting of projections[16,17]. In Alaska, the largest Holocene glacier expansion occurred during the "Little Ice Age" (LIA)[18], AD 1770-1850 AD[18–20], with temperatures 1.3 °C[19] to 2 °C[20] cooler than modern (Supplementary Information Section 3). The geomorphological imprint of this expansion is clearly observable in Sentinel satellite imagery and the fine-resolution ArcticDEM product[21]. Temperature changes during this historical period were of a similar amplitude of forcing (±2 °C) to those projected in the next two centuries[22]. Datasets of glacier change through this historical period, with a measure of uncertainty, are required for understanding the behaviour of ice caps and icefields under a changing climate, and for upskilling projections through data-model comparisons[4,14,16,23,24], but few exist with a long (more than a few decades) and high (decadal) temporal resolution. The aim of this study is therefore to quantify "Little Ice Age" to recent (2020 AD) glacier change at Juneau Icefield (Alaska and British Columbia, Figs. 1, 2a) to determine the rates of, and controls on, icefield-wide shrinkage and thinning.

Juneau Icefield (northern Coast Mountains) straddles the boundary between Alaska (USA) and British Columbia (Canada) (Figs. 1, 2 and Supplementary Fig. 1). It is a temperate, maritime icefield, though the eastern side has a more continental climate[25,26]. The icefield spans an elevation range of 0–2300 m asl, with a large, low-slope accumulation area covering 1400 km$^2$. It comprised 1050 glaciers in 2019[21], covering 3816.3 km$^2$ and with an ice volume of 1051.1 ± 301.7 km$^3$ (calculated using data from ref. 3; Supplementary Fig. 2). Most of the glacier area is held in 40 topographically confined outlet glaciers (2939.1 ± 4.2 km$^2$), which drain directly from the main plateau. Separate to this plateau are smaller ice bodies; 145 valley glaciers (570.9 ± 2.3 km$^2$), 584 mountain glaciers (279.2 ± 6.4 km$^2$) and 281 glacierets (27.2 ± 3.0 km$^2$)[21]. The northern accumulation plateau area lies above 1500 m and has modelled ice thicknesses reaching 420 m$^3$, while the southern plateau is lower, above 1200 m, but ice here is approximately 950 m thick (Supplementary Fig. 1, 2). Taku Glacier has a maximum measured thickness of 1477 m, with a bed up to 600 m below sea level[27]. Icefalls occur on 23 outlet glaciers (including 13 outlet glaciers draining from the main interconnected plateau) and the study region as a whole contains 150 icefalls observed on 55 glaciers[21]. These icefalls have a mean elevation of 1481 m asl (range 810–2121 m) and many occur around the rim of the icefield plateau[21].

Outlet glaciers around Juneau Icefield were at their neoglacial maximum at around 1770 AD (Supplementary Information Section 3)[28–31], with some historical observations of outlet glacier extent also mapped in the early 20th century[32]. Juneau Icefield has an especially long and detailed mass balance record since 1946[25,33–35] and an exceptional database of glaciological observations (e.g., refs. 25,36–42). Positive annual balance was recorded between 1946 to 1949, with high winter cyclonic activity. From 1950 to 1957, annual mass balances were negative, due to decreased cyclonic activity, and below-normal accumulation temperatures[40]. Between 1958 to 1962, warmer winter temperatures and increased precipitation led to increasing annual mass balance. From 1964 to 1975, cool ablation season temperatures and above average winter precipitation caused increased annual balances[40]. Young et al. calculate a rate of −0.57 (−0.11, +0.12) m w. e. a$^{-1}$ for the icefield from 1980–2016, with ice west of the topographic divide undergoing a greater rate of mass loss than the more interior glaciers[43]. In comparison, Berthier et al. calculate a mass balance of −0.65 m yr$^{-1}$ for the Coast mountain range from 1962 to 2006[44]. Glaciers today continue to thin and recede in response to an overall warming[33,45]. From 2000 to 2016, Juneau Icefield had a mass balance of −0.68 ± 0.15 m w.e. a$^{-1}$[46]. Lemon Creek had a cumulative annual mass balance of −1.03 m w.e. from 1990 to 2018, with increasing rates of mass loss, primarily driven by increases in glacier ablation and summer warming[33] (Supplementary Fig. 3d). After decades of glacier advance, Taku Glacier is now receding, with thinning across its entire elevation range between 2013 and 2018, with an average rate of −1.3 m ice equivalent per annum[25,37].

Measured equilibrium line altitudes (ELAs) reach 1499 m asl (mean 2011–2020 AD) on Lemon Creek Glacier and 1159 m asl on Taku Glacier[25,35], with ELAs over 1500 m reported in warm years. The equilibrium line altitude (ELA) at Taku Glacier has risen from 912 m asl (mean 1940–1950 AD) to 1000 m (mean 1986–2018) and to 1159 m asl (mean 2011–2020 AD)[38]. The ELA at Lemon Creek Glacier has risen from 1038 m asl (mean 1961–1970) to 1499 m asl (mean 2011–2020)[25]. This means that the ELA of outlet glaciers is now more frequently reaching the rim of the plateau around the icefield (1200 m asl; cf.[21]). Independent hydrological measurements indicate that glacier ice melt volumes are increasing, with spring glacier melt volume increasing at 16% decade$^{-1}$[47], and an increasing trend in annual glacier ice melt production.

Southern Alaska has a maritime, high-latitude climate regime, characterised by abundant snowfall and cooler summers[33]. Climate in southern Alaska and in the Gulf of Alaska region is largely controlled by the North Pacific Decadal Variability[48]. The dominant modes of the North Pacific Decadal Variability are largely determined by the wintertime strength of the Aleutian Low and sea surface temperature anomalies related to the Pacific Decadal Oscillation.

Precipitation patterns in Juneau are dominated by the Aleutian Low, which is a region of winter-time low pressure near the Aleutian Islands[49]. These low-pressure conditions increase the frequency and intensity of winter storms. A strong Aleutian Low, with a positive Pacific Decadal Oscillation, therefore brings enhanced winter storms and increased precipitation along the Gulf of Alaska[50]. There is strong interannual to multidecadal variability in the Aleutian Low, with a shift in 1976 from mainly negative values (1951–1975) to mainly positive values (1977–2001), leading to increased coastal precipitation[51] and winter snowfall[52]. Since 2015, the Aleutian Low has been weaker[53]. The shift in the Aleutian Low after 1976 is reflected in snowfall received at Juneau; analysis of three-month winter (DJF) precipitation data from the meteorological station at Juneau Airport (NOAA[54]) show mean values of 10,300 ± 1300 mm w.e. (with a 95% confidence interval) from 1951–1975, and 13,200 ± 1200 mm, 1977–2020 AD. A Student's $t$-test showed a statistically significant increase in precipitation between these two time periods ($t$ stat 3.36 > $t$ Critical one-tail 1.67) (Supplementary Fig. 3b). The increase in precipitation occurs mostly in September to November (Supplementary Fig. 3c).

Temperatures from Juneau Airport meteorological station (NOAA[54]; Supplementary Tables 1, 2) are available from 1941-present at a monthly resolution. Mean summer temperatures (1986–2005) reach 13.34 °C, with a mean annual air temperature of 5.67 °C. Winter temperatures show a strong increase in warming, with a trend of 0.35 °C per decade from 1941–2020 ($p < 0.05$). There was an increase in mean winter temperatures of 2.07 °C from 2001–2020 relative to 1941–1970 (Supplementary Table 1), which has lengthened the summer melt season. For mean annual air temperatures, six of the ten warmest years have occurred since the year 2000, and the ten coolest years all occurred prior to 1973 (Supplementary Fig. 3a). Since records began at Juneau Airport, summer temperatures have also increased, with temperatures from 2001 to 2020 0.97 °C warmer than from 1941 to 1970. This trend is in line with observations across Alaska, which shows warming temperatures since the 1970s and exceptionally warm recent years, a shrinking snow-season state-wide, and a longer melt season[55].

This climate shift is also clearly visible in the ERA5 climate reanalysis data[56] (Supplementary Fig. 4). Here, the mean climate

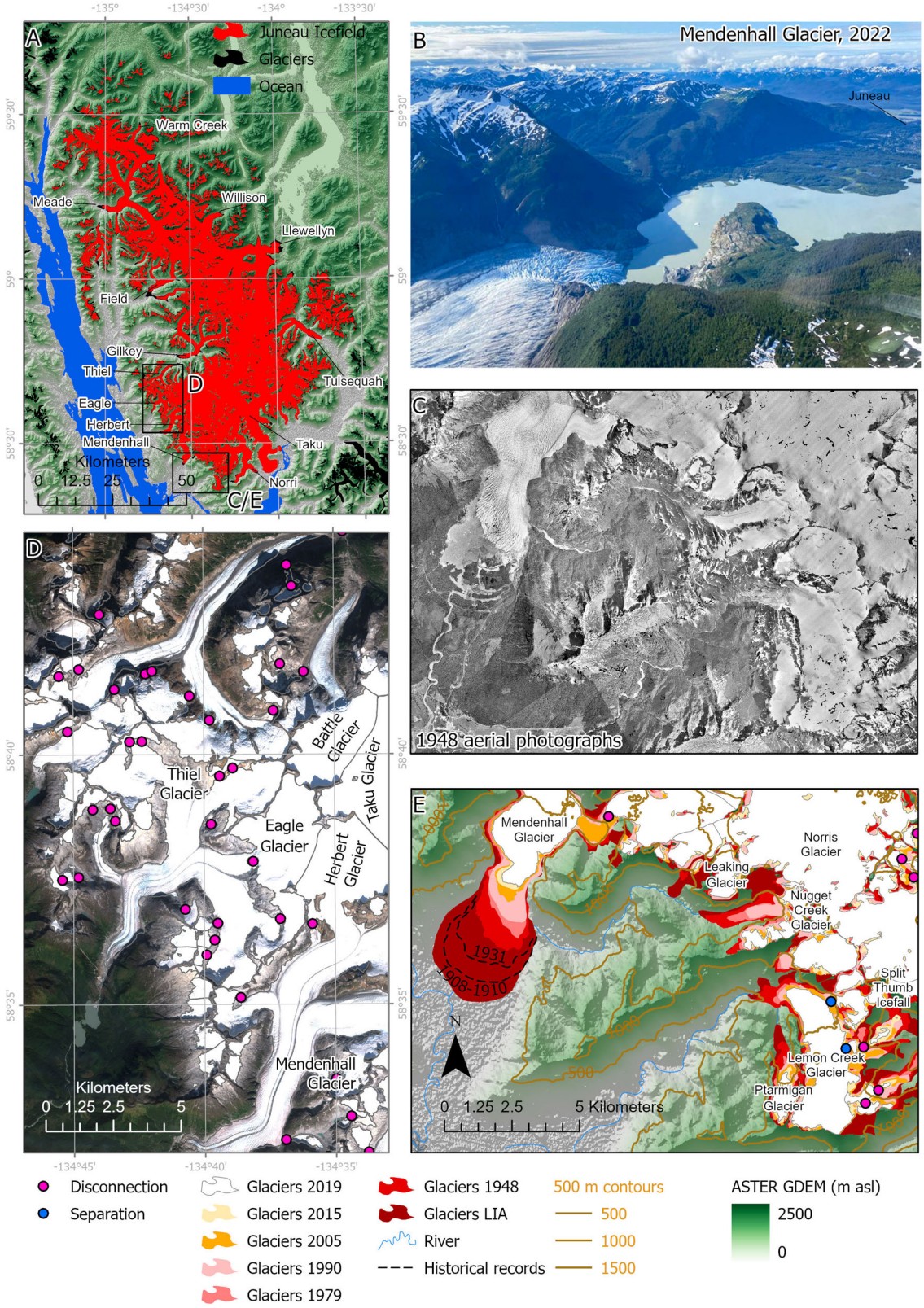

**Fig. 2 | The Juneau Icefield and its evolution over time. A** Overview of Juneau Icefield; glacier extent in 2019 is shown in red. Key glacier names are shown. Glaciers in black not included in the study area. Overlain on background of ASTER GDEM. **B** Field photograph (credit Bethan Davies) of Mendenhall Glacier, in June 2022. **C** Examples of aerial imagery used in glacier reconstruction (1948) for Mendenhall Glacier and Lemon Creek Glacier. Image courtesy of the U.S. Geological Survey. **D** Example of satellite imagery (Sentinel-2A, 2019) for Thiel Glacier (north-flowing) and Eagle Glacier (south-flowing). **E** Example of reconstructed glacier area loss for Mendenhall Glacier and Lemon Creek Glacier. Historical glacier positions from ref. 28. **C**, **E** have the same scale and extent. Extents are shown in (**A**). See also Supplementary Figs. 1, 2. ASTER GDEM is the Global Digital Elevation Model produced by ASTER, courtesy of NASA/JPL-Caltech.

(temperature, precipitation and vertically integrated moisture difference) of 1990–2005 and 2015–2019 are compared with the mean 1950–1980 climate. The post-1976 climate shift is clearly apparent, with warming and increased precipitation in 1990–2005 relative to 1950–1980. However, the difference is much greater for the 2015–2019 period, when temperatures, at both 2 m and 850 mb, are substantially higher. This period sees especially warm temperatures around Juneau (Supplementary Fig. 4e, f) as well as decreased precipitation (Supplementary Fig. 4h).

## Results

### LIA glacial reconstruction

The reconstructed LIA icefield (Figs. 3, 4 and Supplementary Figs. 9–10) covered $5414.95 \pm 111.76$ km²; 29.5% larger than in 2019 AD (Table 1), with reconstructed ablation areas adding $95.26 \pm 17.01$ km³ compared with 2013 AD (Table 2). At this time, Juneau Icefield outlet glaciers extended down onto the plain in several valleys, with many forming large unconstrained piedmont lobes; for example, Hole-in-the-Wall and East and West Twin Glaciers, Mendenhall Glacier, Herbert Glacier, Llewellwyn Glacier and Eagle Glacier (Supplementary Fig. 10). At these piedmont glacier lobes, large concentric moraines represent a substantial period of stability and moraine building for the glacier. In many of these places, the moraines display multiple closely spaced ridge crests. There are some moraines visible between the innermost of these concentric piedmont glacier moraines and the 1948 glacier outlines, for example at Herbert Glacier (Supplementary Figs. 1h, 9), suggesting that the ice may have temporarily stabilized during recession into the valleys, potentially due to changing topographic constraints on the margin. Recessional moraines are also evident at Llewellwyn Glacier, but within the 1948 extent.

For the most part, over-deepenings inside the main moraine crests are infilled with proglacial lake water. Taku and Norris glaciers were the only two glaciers to reach the ocean and to calve directly into a marine fjord. East and West Twin Glacier and Hole-in-the-Wall Glacier drained into an ice-dammed lake, but the remaining piedmont glaciers were land-terminating. Kettle lakes occur frequently within the piedmont glacier moraine crests. The remainder of the outlet glaciers (e.g., Field, Gilkey and Meade glaciers) remained constrained as land-terminating valley glaciers at their most recent neoglacial maximum (Figs. 3, 4, Supplementary Fig. 10). These glaciers terminated on land and had a long, narrow, low-slope glacier snout. Again, as the glaciers have receded from their most recent maximum position, proglacial lakes have formed within the over-deepening, impeding observation of younger recessional moraines.

Taku Glacier shrank by 6.4% (49.95 km²) but was slightly thinner at the terminus during the LIA than today, potentially reflecting its different configuration and marine-terminating environment, compared with the thicker, terrestrially terminating glacier present today[25,37]. Recent recession now sees the formation of lagoons at the ice margin and the onset of a lacustrine or lagoonal terminus. At the LIA, Taku Glacier was coalescent with Norris Glacier in Taku Inlet[28], calving into the fjord (Supplementary Fig. 11). It is likely that this marine-terminating glacier had a long, low-slope, low-elevation snout, as is typical for water-terminating glaciers. We suspect that the glacier was grounded, given the shallow fjord, though this is not constrained. Behind the terminus, we reconstruct an ice-dammed lake at 15 m elevation, following the clear break in slope between the forested mountainside and the low-slope boggy valley floor, as originally mapped by Lawrence, 1950[28]. This is in line with ice-contact fans observed at Twin Glacier Lake moraines and at Wright Glacier terminus further up-valley (Supplementary Fig. 10), which provide additional geomorphological evidence for an ice-dammed lake at 15 m asl. However, little geomorphological evidence of shorelines is visible in the ArcticDEM or satellite imagery. Reconstructing the lake following the 15 m contour generates a lake 95.84 km² in size.

Around the periphery of the main plateau, the LIA extent of valley glaciers and mountain glaciers is generally well constrained by substantial and clear moraines, enclosing ice-scoured bedrock or fluted glacial sedimentary surfaces. There are occasional recessional moraines within these larger moraines. For some glaciers, such as Denver Glacier, limited clear geomorphology visible on satellite imagery precluded reconstruction of the glacier extent, and so the 1948 extent is used as the earliest data source.

Reconstructed geometric equilibrium line altitudes ($ELA_{LIA}$) ranged from 1070 m (Taku Glacier, Lemon Creek Glacier) to 932 m (Mendenhall Glacier) and 1494 m (Llewellyn Glacier) (Supplementary Fig. 10). In comparison, the average ELA for Lemon Creek Glacier from 1998-2019 was 1327 m asl[25,35]. This represents an ELA lowering of ~150 m at Lemon Creek Glacier, relative to today[25,35]. Peripheral glaciers to the east of the icefield had an $ELA_{LIA}$ higher than 1500 m, compared with under 1250 m in the west (Supplementary Fig. 10).

### Overall changes since the "Little Ice Age"

Between the AD 1770 and AD 2019 inventories, 108 glaciers disappeared (Fig. 5), and only 70.53% of glacier area now remains relative to the Neoglacial maximum. 100% of glaciers mapped in 2019 had receded relative to their LIA position, with 47 new ice-contact proglacial lakes developing as glacier termini receded. The largest areal losses (52.5%) were from the Icefield's outlet glaciers (which made up 64.25% of the area during the LIA). Of all the glacier area lost between 1770 and 2019 AD, $538.0 \pm 32.4$ km² was from outlet glaciers. The main outlet glacier termini all receded by ~4 to 5 km, shrinking by up to 69 km² (Supplementary Table 9).

The total cumulative ice loss from across Juneau Icefield is $315.3 \pm 237.5$ km³; this equates to a loss of 24.25% of the ice volume (from ref. 3) between the LIA maximum and 2020 AD. There has been thinning of 150 m at the terminus of Field Glacier, 220 m on Ogive Glacier, 150 m on Gilkey Glacier, and 200 m on Meade Glacier. Bacon Glacier has thinned by 190 m, and Tulsequah by 180 m.

### Accelerating change through the 20th and 21st Centuries

Most glaciers (91%) receded substantially from their LIA extent between 1770 and 1948 AD (Fig. 5), during a period of warming (Fig. 6a). Total glacier area shrank by 12.18% at a rate of 0.07% a$^{-1}$ (3.70 km² a$^{-1}$). This was accompanied by a conservative estimate of volume loss of $0.65 \pm 0.92$ km³ a$^{-1}$ (Fig. 6b–g). Between 1948 and 1979 AD, 342 (30.6%) glaciers advanced, and overall rates of glacier recession were slow (0.12% a$^{-1}$) (Fig. 6b–g; Table 1). This was likely caused by an increase in snowfall relative to the period before 1948 (Supplementary Table 2, Fig. 6f). Most advancing glaciers were either small mountain glaciers or glacierets, likely reflecting their faster response time. Despite these changes, a comparatively low rate of thinning persisted, at an icefield-wide average of 0.11 m a$^{-1}$ (Fig. 6c). Rates of recession before 1948 remain well below those experienced by most outlet glaciers after the year 2005 (Supplementary Table 10).

Slower rates of area loss and thinning continued until 1979, followed by an acceleration of ice loss during the latter 20th and start of the 21st centuries (Table 1). Fewer (8.5%) glaciers advanced between 1990 and 2005 (n = 95). Rates of glacier area loss rose, reaching 0.18% a$^{-1}$ (8.33 km² a$^{-1}$, 1979–1990), and then sharply accelerated to 0.39% a$^{-1}$ (17.54 km² a$^{-1}$, 1990–2005). From 1979–2000, icefield-wide volume loss reached $3.7 \pm 1.6$ km³ a$^{-1}$ (Fig. 6d), also indicating a sharp acceleration from the preceding periods, with a mean thinning of $-0.48$ m a$^{-1}$ across all glaciers (Fig. 6c). Glacier tongues thinned across the icefield (Fig. 7), with thinning reaching higher elevations up-glacier than before (Fig. 7). Thinning reached elevations of 1380 m asl on Mendenhall Glacier, 1490 m for Tulsequah Glacier, and the plateau at 1550 m for Meade Glacier.

This rate of ice loss continued into the 21st century. Between 2005–2015 AD, 99% of glaciers receded, and 10% of glaciers

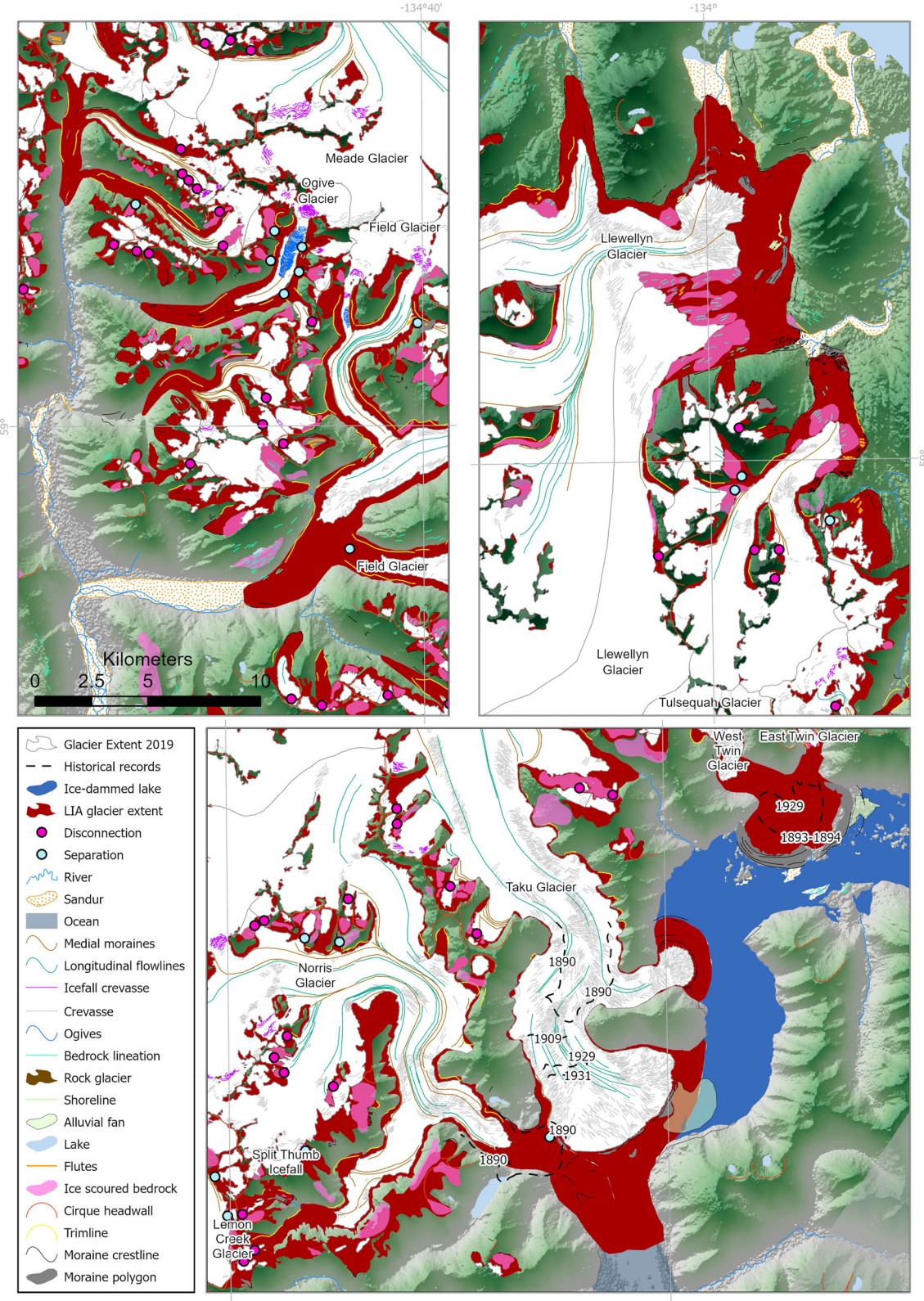

**Fig. 3 | Details and examples of the geomorphological data and icefield reconstruction for the "Little Ice Age" (LIA) maximum.** Geomorphological data, including all shapefiles, are available from ref. 21. Overlain on ASTER GDEM, the Global Digital Elevation Model produced by ASTER, courtesy of NASA/JPL-Caltech.

disappeared between 2005 and 2019 AD. From 2005 onwards, glacier areal loss accelerated sharply relative to previous time periods, reaching 0.60% $a^{-1}$ (25.36 km² $a^{-1}$, 2005–2015) and then 0.96% $a^{-1}$ (38.47 km² $a^{-1}$, 2015–2019) (Fig. 6b–g). Rates of recession were 1.5 times faster from 2015–2019 (Mean (M) = 5.88, Standard Deviation (SD) = 5.26) than from 2005–2015 (M = 2.48, SD = 2.11), t(1097), p < 0.001.

Icefield outlet glaciers experienced severe volume loss after 2010 (Fig. 6c, g; Table 2), with thinning across the icefield plateau at

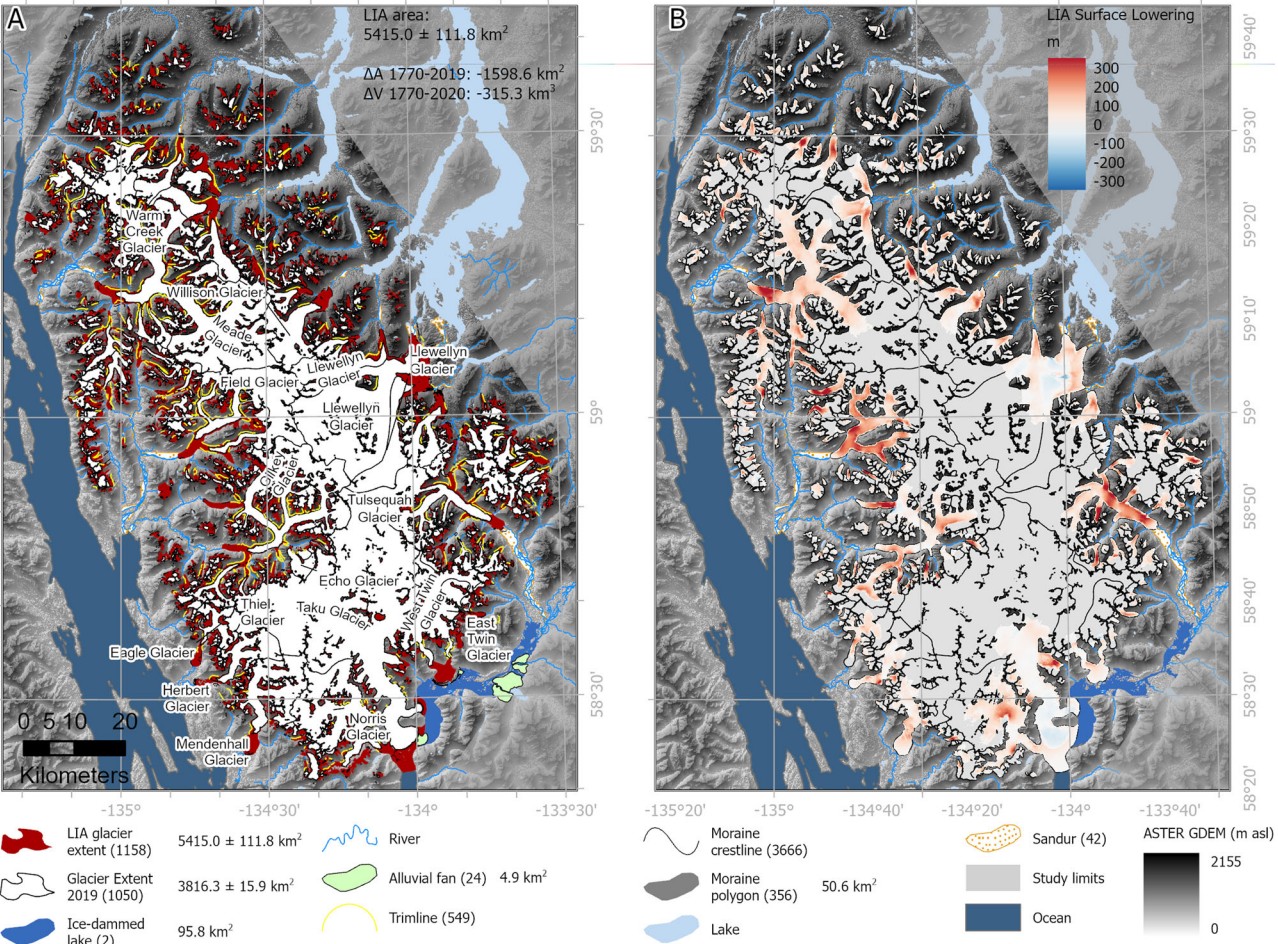

**Fig. 4 | "Little Ice Age" (LIA) reconstruction of Juneau Icefield. A** Reconstruction of Juneau Icefield during the Little Ice Age. Geomorphic data from ref. 21. Legend shows number of features in brackets. **B** ice surface lowering (m), LIA (1770) to present (year 2013; comparison with Copernicus DEM). More information on the LIA reconstruction is available in Supplementary Information with glacier equilibrium line altitudes (ELAs) from the LIA shown in Supplementary Figs. 9–11. Feature counts are shown in brackets in the legend. ASTER GDEM is the Global Digital Elevation Model produced by ASTER, courtesy of NASA/JPL-Caltech.

elevations of up to 1800 m asl (Fig. 7) and volume loss at a rate of $5.91 \pm 0.8 \ km^3 \ a^{-1}$ (2010–2020). The whole icefield loss equated to 5% of the total volume reconstructed in 2017–18 by ref. 3 from 2010–2020. Alongside this thinning we observe increased glacier fragmentation, with both terminus separation and glacier disconnections (*sensu* ref. 21), where glacier tributaries are detached from their accumulation areas, occurring increasingly frequently (Fig. 6h, i).

Overall, rates of glacier areal recession rose slowly from 1948–2005, followed by a sharp acceleration in rates of glacier area loss after 2005. Icefield-wide, rates of glacier area shrinkage were 7 times faster from 2015–2019 relative to 1948–1979, and 5 times faster relative to 1979–1990 (Table 1, Fig. 6b–g). There are three distinct periods in icefield volume loss, with relatively consistent rates from LIA-1979, followed by a rise from 1979–2010, and a doubling of rates of volume loss after 2010 (Fig. 6d). There has thus been a steep acceleration in the rate of ice mass loss across the icefield since the early twenty-first century.

**Snowline and Albedo change**

This rapid glacier recession was accompanied by ELAs rising to the plateau height in the period 2005–2020[25,35] (Fig. 6i), and exceptionally high snowlines on the plateau in 2018[38] and 2019 (Figs. 5, 6k, Supplementary Table 13; Supplementary Fig. 14). Late summer snowlines mapped in 2019 from Sentinel-2 imagery (Figs. 5, 6k) had a mean altitude of 1612 m asl (SD 162 m; n = 178). Snowlines on Taku Glacier

were a mean altitude of 1445 m in 2019, compared with a measured ELA in 2019 of 1528 m[35]. The snowline here is therefore well above the southern plateau at 1200 m asl. No snow was observed in the Sentinel-2 imagery on Lemon Creek Glacier at the end of the ablation season in 2019, in line with the measured ELA of 2023 m that year[35]. Snowlines on Meade, Llewellyn and Field Glacier had a mean elevation of 1671 m (SD 119 m), well above the height of the northern plateau at 1500 m asl (Fig. 5). This icefield-wide trend has continued in subsequent years (Fig. 6k, Supplementary Tables 13, 14 and Supplementary Figs. 13, 14). These datasets show that, although snowline elevation was unusually high in 2019, snowlines are reaching the plateau frequently, with both snowlines and ELAs regularly occurring above the plateau height of 1200 m since 2005 and increasingly frequently in the last decade of the study period.

Alongside this trend of rising snowlines, we also observe a decrease in icefield wide average albedo (Fig. 6l, Supplementary Tables 15–17). When clipped to the 1990 glacier outlines, and thus taking into account terminus recession and icefield fragmentation as well as the impact of darkening snow and ice surfaces, the mean albedo from 1987–2009 was $0.81 \pm 0.03$ (95% confidence interval). From 2010 to 2023, this decreased to $0.67 \pm 0.03$. The albedo of the plateau above 1500 m (compared with the Copernicus DEM) also decreased from $0.92 \pm 0.02$ (1987–2009) to $0.78 \pm 0.04$ (2010–2023). The months of September and August in 2018 and 2019, which were characterised by exceptionally high snowlines, likewise had exceptionally low icefield-

**Table 1 | Total summed areas and rates of change across Juneau Icefield, with climate data from each period**

| Year | Total glacier area (km²) | Mean glacier area (SD), km² | Count of glaciers | Δ Area (km²) | % change | Total rate of glacier ΔA(km² a⁻¹) | Rate of glacier ΔA (% a⁻¹) | Mean rate of glacier change (km² a⁻¹) | Mean % change per annum | No. Advancing | Median no. years (SD) | Summer temperature anomaly (°C) |
|---|---|---|---|---|---|---|---|---|---|---|---|---|
| 1770 | 5415.0 ± 111.8 | 4.7 (32.4) | 1158 | | | | | | | | | |
| 1948 | 4755.5 ± 26.0 | 4.1 (31.0) | 118 | −659.0 ± 137.7 | −12.18 | −3.70 | −0.068 | −0.003 ± 0.001 | 0.20 | 67 | 178.0 (0) | −0.83 |
| 1979 | 4584.0 ± 97.6 | 34.0 (30.7) | 118 | −171.6 ± 123.6 | −3.61 | −5.48 | −0.115 | −0.007 ± 0.002 | −0.03 | 342 | 31.0 (14.3) | −1.02 |
| 1990 | 4496.1 ± 55.8 | 3.9 (30.4) | 117 | −87.9 ± 153.4 | −1.92 | −8.33 | −0.182 | −0.007 ± 0.002 | 0.18 | 387 | 11.1 (1.0) | −0.30 |
| 2005 | 4238.7 ± 47.7 | 3.7 (29.8) | 114 | −258.8 ± 103.5 | −5.76 | −17.37 | −0.390 | −0.015 ± 0.003 | 1.14 | 95 | 14.9 (0.3) | 0.01 |
| 2015 | 4001.5 ± 69.4 | 3.5 (29.3) | 1102 | −235.8 ± 117.1 | −5.56 | −25.76 | −0.598 | −0.022 ± 0.005 | 2.54 | 0 | 9.0 (0.5) | −0.40 |
| 2019 | 3816.3 ± 15.9 | 3.3 (28.9) | 1050 | −185.1 ± 85.3 | −4.63 | −38.47 | −0.961 | −0.033 ± 0.006 | 5.88 | 5 | 5.0 (0.4) | 0.55 |
| Total change | | | −108 | −1598.6 ± 127.6 | | | | | | | | |

"Advancing" is the number of glaciers with increased glacier area. Mean rate of glacier area change (km² a⁻¹) is shown with the 95% confidence interval. Summer temperature anomaly is compared with the 1986–2005 AD mean. SD standard deviation.

wide mean albedos of 0.44 to 0.48, respectively (LC08 sensor). In contrast, the lower icefield average snowline in August 2021 (Supplementary Table 16) is reflected with a higher albedo of 0.66 in the LC08 sensor.

When comparing the time periods 1987–2009 and 2010–2023, the icefield-wide albedo within the 2019 glacier outlines was lower from 2010–2013 (mean = 0.71, standard deviation = 0.14) than from 1987–2009 (mean = 0.84, standard deviation = 0.08); $t$ (90) = 5.48, $p < 0.05$. This decrease is caused by more pixels with a lower albedo in summer, especially on the plateau; here, a regression for average albedo over time results in an $r^2$ value of 0.23 ($p < 0.05$). Many bright pixels with high albedo however remain throughout the study period, especially at higher altitudes on the plateau.

### Icefield fragmentation

The recession and thinning of Juneau Icefield glaciers has resulted in increasing icefield fragmentation. This includes the separation of glacier tongues in the valleys, and disconnections higher up the glacier, where bare rock appears within the glacier polygon as a result of thinning, usually over thin, steep and heavily crevassed ice such as in icefalls[21]. While 291 disconnections were first mapped in ref. 21, which were frequently observed in association with icefalls, here we calculate the timing of disconnection by comparing the mapped disconnections with the new glacier outlines, and analyse the thinning occurring at the site of the disconnections. The mean elevation of the disconnections has remained relatively constant over time (Fig. 6i), with a total mean elevation of 1299 m asl (1362 m in disconnections after 2005); this is controlled by the location of steep ice around the rim of the plateau[21]. This altitudinal range (1200–1400 m asl) was subjected to thinning in the period 2010–2020 (Fig. 7). For example, in a subsidiary accumulation basin on Gilkey Glacier, thinning of 1.4 m a⁻¹ is observed at a height of 1340 m adjacent to a glacier disconnection. At Thiel Glacier, thinning of 1.7 m a⁻¹ is observed at an altitude of 1034 m adjacent to the site of a pre-2019 disconnection.

These disconnections are occurring at a uniform altitude and slope through time (Fig. 6i) but are occurring more frequently after 2005 AD as glacier ELAs increasingly intersect this altitudinal range (Fig. 6h; refs. 21,35) and thinning occurs at higher elevations (Fig. 7). Glacier snowlines in 2019 AD were above the elevation of icefalls around the rim of the plateau[21] (Fig. 5), resulting in net ablation at the height of many of the icefalls. Glacier tongues down-stream of glacier disconnections show increasing debris cover and thinning. For example, thinning of up to 9 m a⁻¹ on Thiel Glacier tongue, with a glacier-wide mean of −3.33 m a⁻¹, is far above the mean of 1.3 m a⁻¹ for outlet glaciers (2010–2020, Fig. 6c).

## Discussion

This Juneau Icefield-wide examination of glacier and climate change over this 250-year time period reveals that rates of glacier recession have sharply accelerated since 2005, relative to earlier time periods (Fig. 6b, g). Observed rates of glacier volume and area loss are rapidly accelerating, demonstrating a threshold response to a warming climate, a rising ELA and snowlines, and decreasing regional albedos, in contrast to published modelling studies, which suggest a linear rate of volume loss to 2040 AD and acceleration only after 2070 AD[6,14]. In summary, temperatures rose by 1.39 °C at Juneau Airport (1941–2020, Figs. 5 and 6f), with a shift occurring in the 1970s, in line with regional observations across Alaska[55]. This is related to a shift to mainly positive values in the Pacific Decadal Oscillation from 1976[51], bringing increased precipitation and warmer temperatures to Alaska[50,51] and to Juneau (Supplementary Tables 1, 2; Supplementary Figs. 3, 4). Temperatures were relatively stable from 1990–2005 (Fig. 6f). Icefield thinning and area loss remained stable during this time period (Fig. 6b–d). From 2010–2020, there was another rise in temperature, and in ELA[25,33,35], with the ELA and late-summer snowlines now intersecting with the

**Table 2 | Total volume change for Juneau Icefield, LIA to 2020 AD**

| Time period | Volume Change | | | Mean glacier area-averaged dh/dt (m) ± 95% confidence interval[a] | | | | |
|---|---|---|---|---|---|---|---|---|
| | ΔV (km³ a⁻¹) | Total volume change (km³) | Cumulative volume change (km³) | All glaciers | Glacierets | Mountain Glaciers | Valley Glaciers | Outlet Glaciers |
| 1770–2013[b] | −0.4 ± 0.1 | −95.26 ± 17.01 | | | | | | |
| 1770–1948 | −0.7 ± 0.9 | −115.9 ± 163.4 | −115.9 | −0.11 ± 0.01 | −0.12 ± 0.03 | −0.12 ± 0.01 | −0.11 ± 0.03 | −0.10 ± 0.02 |
| 1650–1948 | −0.4 ± 0.6 | | | | | | | |
| 1880–1948 | −1.7 ± 2.4 | | | | | | | |
| 1948–1979 | −1.0 ± 0.7 | −31.3 ± 23.0 | −147.2 | −0.28 ± 0.03 | −0.17 ± 0.06 | −0.28 ± 0.03 | −0.44 ± 0.09 | −0.33 ± 0.11 |
| 1979–2000 | −3.7 ± 1.6 | −78.2 ± 33.0 | −225.4 | −0.51 ± 0.02 | −0.37 ± 0.05 | −0.49 ± 0.03 | −0.73 ± 0.07 | −0.69 ± 0.14 |
| 2000–2010 | −3.1 ± 1.0 | −30.8 ± 10.1 | −256.2 | −0.21 ± 0.04 | −0.10 ± 0.07 | −0.17 ± 0.05 | −0.50 ± 0.09 | −0.45 ± 0.22 |
| 2010–2020 | −5.9 ± 0.8 | −59.1 ± 8.0 | −315.3 | −0.74 ± 0.03 | −0.58 ± 0.05 | −0.69 ± 0.03 | −1.10 ± 0.09 | −1.21 ± 0.21 |

[a]Mean dh/dt is the glacier area-averaged thinning, averaged for all glaciers, provided with 95% confidence interval. Glaciers with no data coverage are excluded. The low spread in the data results in a small value for the uncertainty in the mean.
[b]Change from LIA ("Little Ice Age") to 2013 is the difference between the reconstructed LIA ice surface (ablation areas only) and the Copernicus DEM, with an approximate date of 2013 AD. Note that this is only for the area of the glacier ablation areas (shown in Fig. 2).

icefield plateau (Figs. 5, 6I; ref. 38), driving lower albedos and widespread thinning across the plateau (Fig. 7), with a thinning rate 1.9 times higher than from 1979–2000, and a sharp and substantial increase in rates of glacier recession. Glaciers receded 2.2 times faster in 2015–2019 relative to 1990–2005, with an increase in mean summer temperature anomaly of just 0.55 °C (2015–2019, from NOAA data[54], Fig. 6f, j, I; Supplementary Tables 1, 2) relative to the 1986–2006 mean.

The critical driver of the accelerating glacier mass decline at Juneau Icefield is that the rising ELA (mean 1437 m asl at Lemon Creek Glacier, and 1188.5 m at Taku Glacier, with highs of 2023 and 1528 m respectively, 2011–2020 AD[35]) and late summer snowline now intersects the elevations of the icefield plateau (1200 m to 1500 m) (Figs. 5, 6i), driving thinning across the plateau (Fig. 8) and increased glacier fragmentation at points of key vulnerabilities (i.e., at sites of steep, heavily crevassed ice). This rising ELA drives a reduced accumulation area ratio (cf. ref. 25). Icefield hypsometry then creates conditions prone to a strong mass balance response to moderate climate forcing[8,12], because the low-slope icefield plateau is susceptible to significant loss of accumulation area. In coming decades, this will drive further decrease of the icefield's accumulation area, and will continue to drive accelerated glacier mass change.

We hypothesise that the dramatic reduction in icefield accumulation area is decreasing overall icefield albedo. As snow cover duration and extent is reduced (cf. Fig. 5), darker rock, firn and ice surfaces are increasingly exposed, which absorb more solar radiation, enhancing the ablation rate. This albedo feedback (cf.[15,57] and illustrated in Fig. 8) is likely to then contribute to further icefield thinning. This effect is compounded by deposition of black carbon[58] and dust[59], which is set to increase in future, due to a rise in tourism and wildfires[60].

Thinning on low-slope icefield plateaux, as observed in Fig. 7, is likely to drive an elevation-SMB feedback by lowering the icefield surface and driving increased melt in warmer air temperatures (due to the lapse-rate induced rise in air temperature at lower elevations[10,11,61]) (Fig. 8). This response is fundamentally unstable, as thinning drives increasingly negative mass balances, even in the absence of further climate forcing[61]. In some places across the Juneau plateau, the ice is over 600 m thick[3] (Supplementary Fig. 2), so the ice-elevation feedback has a large range over which to operate. This topographic feedback can lead to an increased frequency of extreme negative mass balance rates, especially when the climate data diverges from the mean[11]. Once initiated, long-term thinning is inevitable, because even in the event of climate stabilisation, the mass balance-elevation feedback will inhibit glacier regrowth (cf.[14]), likely forming an irreversible tipping point in glacier dynamics.

Thinning in vulnerable areas of thin, heavily crevassed ice such as the icefalls that surround the plateau is encouraging icefield fragmentation and disconnection of glacier accumulation and ablation areas[8,9,21,62,63]. Bare rock appearing within the glacier boundaries indicates reduced nourishment between the accumulation basin and glacier outlet tongues and is likely to enhance melt through radiative forcing, encouraging a positive melt feedback[15,62]. Disconnected glaciers stagnate and downwaste once detached in this manner[62]. Thin debris accumulates on the ice surface, darkening the surface and thereby enhancing melt. In some places, debris accumulates to the point that the glacier becomes debris-covered (>50% of the tongue is covered in thin debris[21]). Thicker debris cover on glacier ice encourages in situ down-wasting but slows rates of glacier area loss due to the insulating impact of debris cover on surface melt[64]. The proportion of debris-covered glacier area to clean-ice glacier area will likely increase as glacier shrinkage continues, increasing the relative importance of debris cover. We hypothesise that increased bare rock exposure and the loss of ice within and around glaciers will act to darken the region, resulting in the observed reduced regional albedo, which may contribute to further local warming. Finally, structural mapping across Juneau Icefield indicates areas of weakness associated with steep icefalls[21]. Their abundance indicates that further disconnections, fragmentation, and downwasting are inevitable[21]. Further thinning will drive the development of new icefalls across topographic steps, as the ice surface becomes increasingly influenced by bed topography. Increased glacier disconnection at icefalls and subsequent downwasting of remnant glacier tongues is therefore expected to become more frequent in future decades, and will form a hysteresis behaviour, whereby glacier regrowth is inhibited.

These processes may be exacerbated as further increases in temperature alter the ratio between solid and liquid precipitation across Juneau Icefield, reducing the amount of snowfall and decreasing accumulation. Top-heavy icefield glaciers are predicted to experience the most significant losses of glacier area over coming decades[8], and the accelerating trends observed in the recent past at Juneau Icefield are likely to continue. The hypsometrically controlled melt-accelerating feedbacks will contribute to a hysteresis in the system, impeding glacier regrowth even in the event of climate cooling.

The acceleration in mass change observed at Juneau Icefield has significant implications for understanding how other icefields may behave in our warming world. Significantly, Alaska is not only a major contributor to ongoing sea level rise, but also contains some of the world's largest plateau icefields (Fig. 1), which will also be susceptible to rising ELAs and SMB and altitude feedbacks. Equilibrium line altitudes are already approaching the plateau for Sargent Icefield and Bagley Icefield. Substantial low-slope icefields are also present in Canada (Devon Island Ice Cap[65], Agassiz Ice Cap, Penny Ice Cap), Greenland (Hans Tausen Iskappe[10]) and Norway (Svartisen,

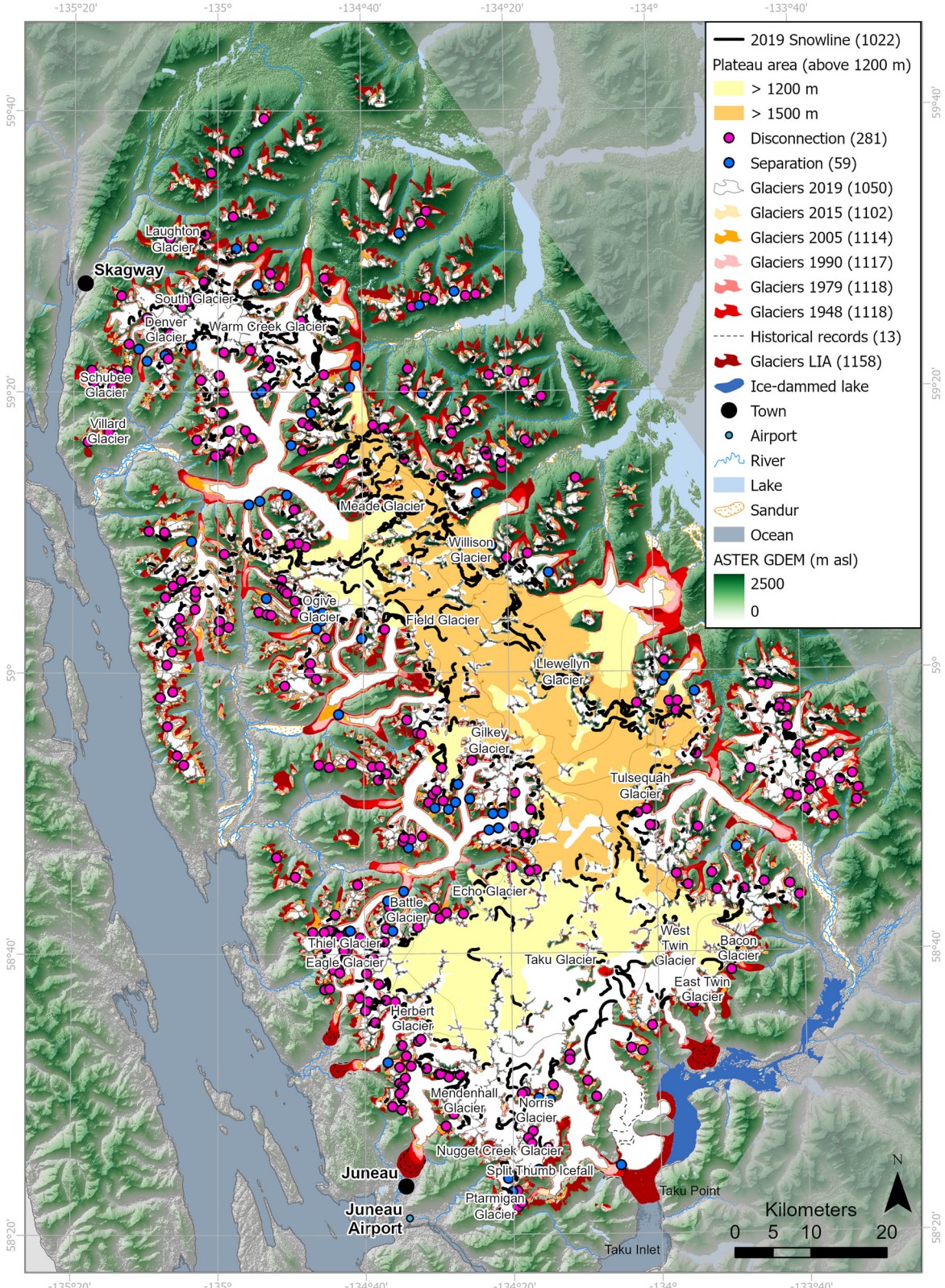

**Fig. 5 | Glacier area change for Juneau Icefield, LIA ("Little Ice Age") to 2019 AD.** Number of glaciers in each time slice is shown in the legend in brackets. Late summer snowline for 2019 AD is shown. Historical recorded glacier extents are available for Eagle Glacier, Herbert Glacier, Mendenhall Glacier, Taku Glacier and East and West Twin Glaciers[28,96,97] (see Supplementary Information Section 3.2). Overlain on ASTER GDEM, the Global Digital Elevation Model produced by ASTER, courtesy of NASA/JPL-Caltech.

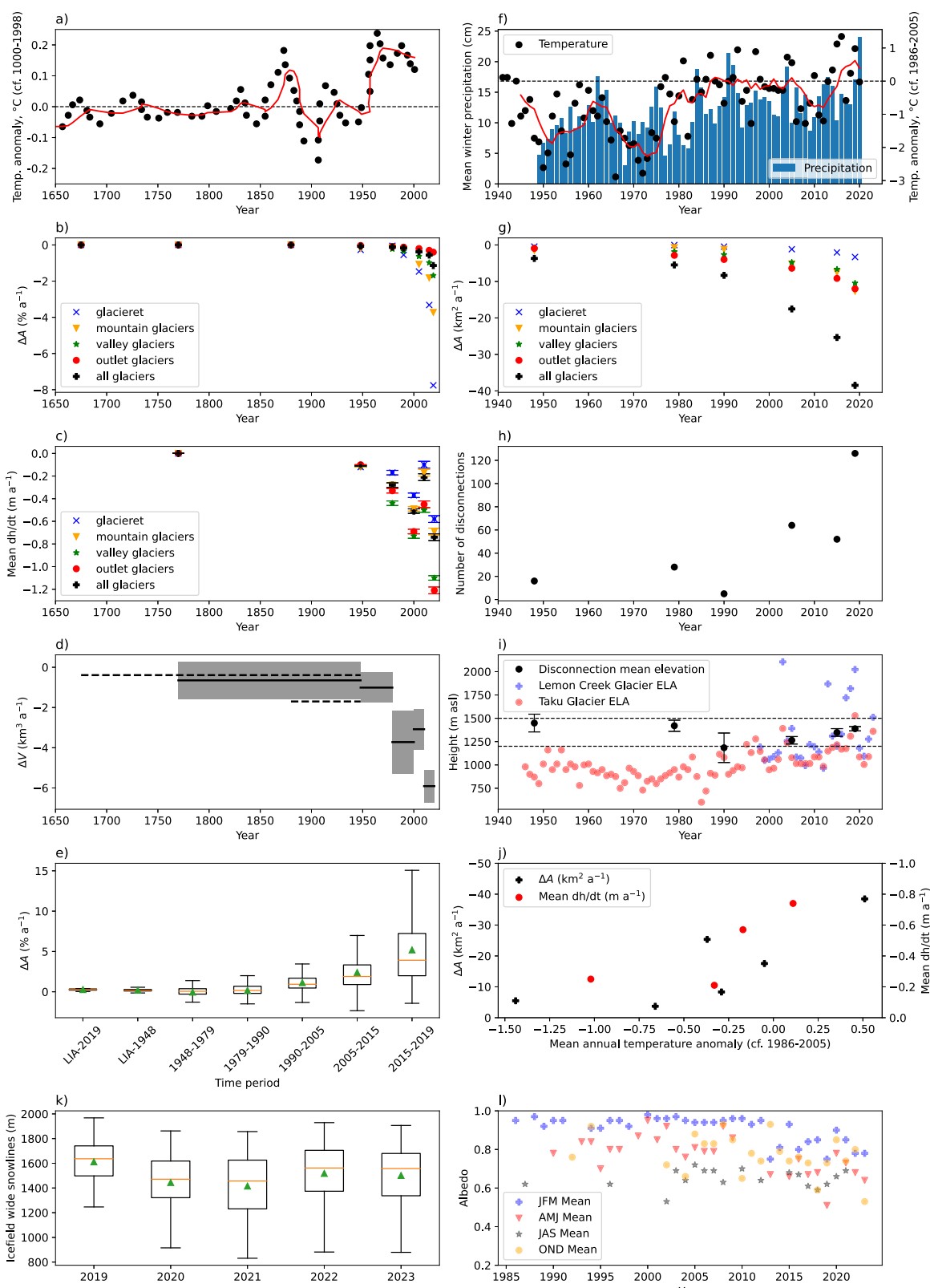

Jostedalsbreen). Plateau ice masses are also important along the Antarctic Peninsula and on James Ross Island. Glacier disconnections may also be critical in bottom-heavy plateau icefields with substantial low-lying outlet glaciers, such as Kluane, Yakutat, Brady and Bagley icefields (cf.[8]; Fig. 1). Conversely, loss of low-lying ablation regions may in some situations have a stabilising impact, because as the low-lying high-ablation areas are lost, this could lead to less negative specific

mass balances[14], emphasising how these complex processes can be icefield-specific. However, this stabilising influence will not offset the pervasive effects of the SMB, albedo and altitude feedbacks if ELAs reach the plateau.

By examining the observed centennial changes in an Alaskan plateau icefield to climate, we show that glacier recession and volume loss have accelerated dramatically in the last 20 years, with marked and

**Fig. 6 | Summary plots of climate and glacier area and volume change.** Time frames (x axes) are consistent in both columns to allow comparison of climate, volume and area change through time. **a** Summer air temperature anomalies from varved proglacial lake sediments in southern Alaska[98]. Red line is the 5-year moving average. Anomaly is computed for the year 1000–1998 AD. **b** Glacier area change through time (% a[−1]) for different types of glaciers. **c** Glacier area-averaged thinning through time (dh/dt) for different classes of glacier. The mean dh/dt (m a[−1]) is calculated for each glacier, and these values are then averaged for each glacier class for each time period. 95% confidence interval is shown. **d** Total glacier volume change, with uncertainties. Dashed lines show rates for an earlier and a later LIA ("Little Ice Age") maximum, respectively. **e** Boxplots of annualised rates of glacier recession for all glaciers where data exist. Mean is shown as green triangle and median as an orange line. **f** Temperature anomaly (scatter plot with 5 year moving average, red line), compared with the 1986–2005 mean, from Juneau airport weather station[54] (see Fig. 5), and mean winter precipitation (bars) as measured at Juneau airport meteorological station. **g** Glacier total summed area change (km[2] a[−1]), focused on the time period 1940–2020 AD. **h** Number of glacier disconnections observed in each time slice. **i** Glacier equilibrium line altitudes from Lemon Creek Glacier and Taku Glacier. Derived from the USGS Benchmark Glacier Programme[35]; see also reference publications[25,33]. Mean elevation of observed disconnections, with error bars of one standard deviation, also shown. Dashed lines indicate the plateau at 1200 m and 1500 m, respectively (refer to Fig. 5 for the mapped plateau area). **j** Temperature and mean rate of glacier recession (km[2] a[−1]; black) and glacier thinning (dh/dt (m a[−1]); red). **k** Boxplots of mapped snowlines, 2019–2023. Further information in Supplementary Tables 13–14 and Supplementary Figs. 13 and 14. **l** Seasonal means each year for albedo for Juneau Icefield. Further data available in Supplementary Tables 15–18. Source data are provided as a Source Data file.

accelerating thinning and recession, compared with the rest of the twentieth century. This accelerated mass change is induced by a number of hypsometrically controlled melt-accelerating feedbacks that drive accelerated thinning and volume across the plateau. The destabilising surface mass balance-elevation feedback, loss of accumulation area, albedo feedback, and glacier disconnections associated with plateau icefields are therefore already occurring here.

Alaska is the largest regional sea-level contributor from glaciers in GlacierMIP[5], with a linear decrease in glacier area and volume predicted to the year 2100 AD, with high uncertainty in rates of mass loss for any given climate scenario, due to the high ice volume in this region. However, once ELAs rise and thinning occurs on the plateau, accelerated mass change occurs due to a series of feedbacks that accelerate melt. Other large icefields and icecaps in Alaska (such as the Harding, Bagley and Kluane icefields, Fig. 1) and the Arctic (e.g., Devon Ice Cap, Agassiz Ice Cap, Penny Ice Cap, Hans Tausen Iskappe), which also occur at low elevations and have similar low-slope plateau accumulation areas, are likely to also display an accelerating rate of future volume loss. Therefore, the outputs of existing large scale regional to global glacier model projects and ensembles should be interpreted with care, and we suggest that more historical data are integrated into future simulations through data-model comparisons to ensure that melt-accelerating behaviours are anticipated, evaluated, and accounted for. The results of this work could have large implications given the important sea-level contributions of Alaska, Canada and Greenland's icefields and the important water resources provided by plateau icefields in the Rocky Mountains (e.g., Columbia Icefield), which could all be subject to similar processes as snowlines rise under ongoing climate warming and these major icefields reach regional tipping points in mass change and volume loss.

## Methods

Baseline glacier outlines were derived from the Randolph Glacier Inventory (RGI) v. 6.0[66,67]. We used glacial geomorphological mapping (ref. 21 and reviewed in Supplementary Information Sections 3 and 5) and historical records[28] to reconstruct LIA extent and then volume[68,69]. We then applied structure-from-motion algorithms in MicMac[70,71] to 20th Century archival aerial photographs (1948 and 1979 AD) as well as topographic maps to reconstruct glacier area and surface elevation in 1948 and 1979 (see Supplementary Fig. 6 for data coverage). Satellite imagery was also used in 1990, 2005, 2015 and 2019 to altogether yield glacier outlines from 1770, 1948, 1979, 1990, 2005, 2015 and 2019 AD (Supplementary Fig. 7), and original icefield surface DEMs at -1770, 1948, and 1979 AD. We combined these with datasets of ice surface elevation derived from structure-from-motion algorithms using ASTER imagery from 2000, 2010 and 2020[72].

### Remote sensing datasets

Glaciers in 2019 AD, glacial structures, glacial geomorphology and lakes were used from ref. 21. These features were mapped in ESRI ArcGIS (projection WGS 84, UTM zone 8 N), using 10 m resolution Sentinel-2 imagery (swath 290 km), and these data and methods are published in full in ref. 21; methods and datasets are summarised here for completeness. Late-summer satellite images with limited cloud cover and snow on the glacier ablation regions were chosen. Landsat imagery was used to map glacier extent in 2015 (Landsat 8), 2005 (Landsat 7 ETM+), 1990 (Landsat 4–5 MSS), and 1980–1982 (using a combination of archival aerial photographs and Landsat 3 MSS imagery where there was no coverage). Landsat imagery was acquired pre-registered to Universal Transverse Mercator (UTM) World Geodetic System 1984 ellipsoidal elevation (WGS84) zone 8 N projection. Image resolution ranged from 4 m (aerial orthomosaics) to 10 m (Sentinel-2), and 30–80 m (Landsat).

Stereo aerial photographs from the 1979 Alaska High Altitude Photography (AHAP) survey were used to map glacier extent and structures from 11th–12th August 1979, especially in areas where significant glacier changes have occurred. This campaign acquired black and white photography at 1:120,000 scale resolution. Cloud cover is minimal and snow patches are limited in area and scope. These photographs were scanned by the USGS EROS Data Centre at 25 microns (1200 dpi).

Aerial stereo photographs from the 1948 aerial photography survey of Alaska were used to map glacier area. Photographs were acquired on 13th August 1948. USGS topographic maps dating from 1948 were used to supplement the 1948 aerial photographs, where the orthomosaic was poor, or where original aerial photographs were unavailable on the USGS server (Supplementary Fig. 6). This was mostly the case in Canada. Here, USGS maps (AK_A-tlin_360559_1960_250000) were used, where the topographic maps were deemed sufficiently high in quality (see Supplementary Tables 3, 5), with a good correspondence between mapped lakes, rivers and topography. Where topographic maps showed a low correspondence, the 1979 AHAP mosaics were used, and, where these were also unavailable (in the eastern-most parts of the study area), Landsat imagery was used (Supplementary Fig. 6). Overall, 66.6% of the glacier area was mapped from the 1948 ortho mosaics, and 27.8% of glacier area was mapped from topographic maps (Supplementary Table 3). This means that the glacier extent in 1948 is the best available estimate given the data availability. Annual rates of recession are calculated using the date of the imagery or data source used to map glacier extent. The glacier extent in 1948 is considered a minimum; use of later datasets results in some glaciers being underestimated due to continued glacier recession.

Topographic data were derived from the 2 m resolution Arctic-DEM v3.0 release 7[73], except where tiles were missing, and ASTER GDEM[74], ALOS DEM[75] and novel DEMs derived from the aerial photographs using structure-from-motion methods. A derived composite DEM is mostly ArcticDEM, with data gaps being filled with a void-filled 30 m resolution ASTERGDEM v2.0[74]. For this process, both DEMs were projected in UTM Zone 8 N, and clipped to the same extent. All

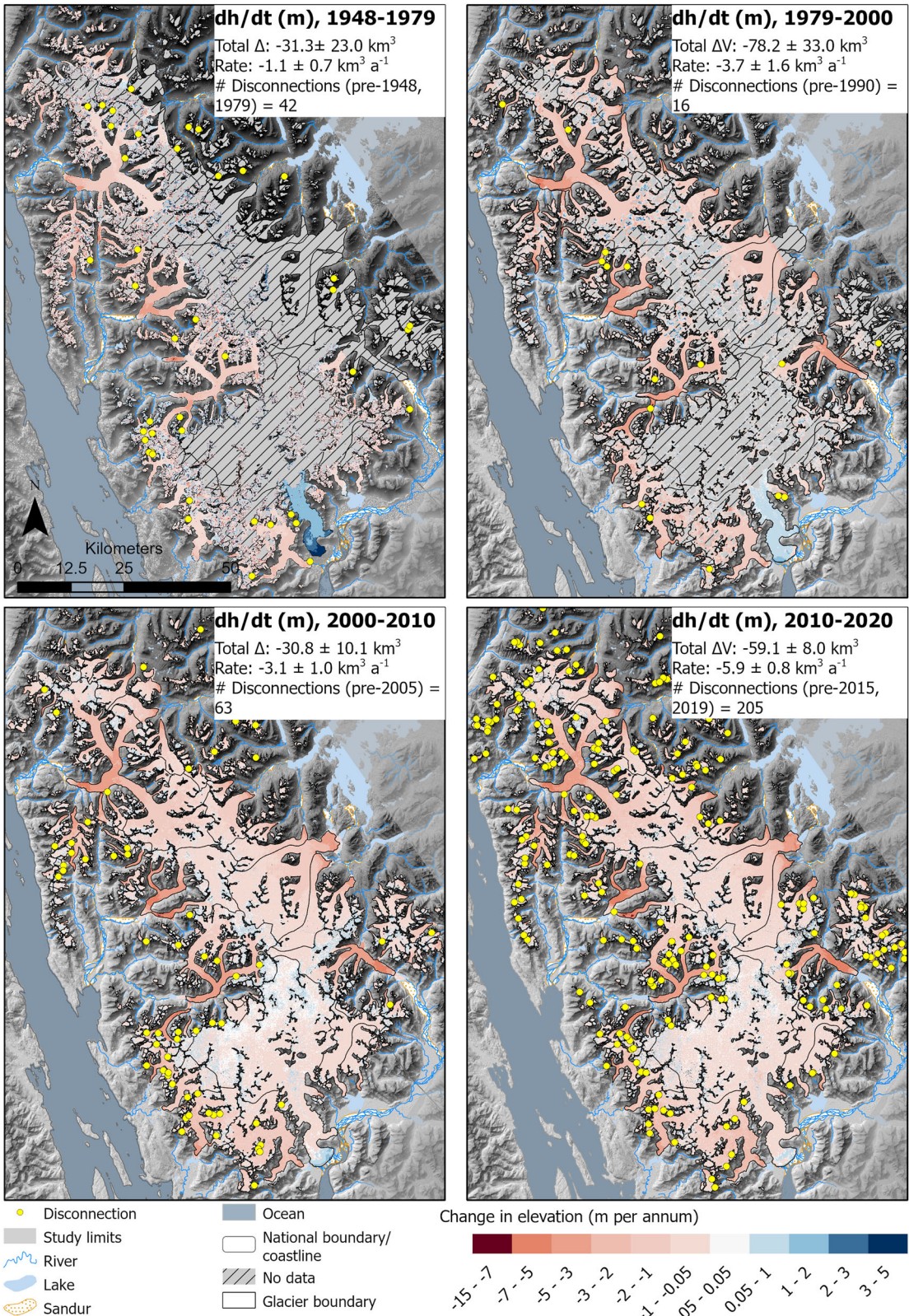

**Fig. 7 | Glacier surface elevation change per year (dh/dt) for each glacier in each time period. Values are given in metres.** Yellow circles are the glacier disconnections observed within each timeslice. Zero elevation change shows as colourless; negative change (thinning) as red and positive change (thickening) as blue. Number of disconnections is shown in each panel; however, as date of disconnection is taken from the area change datasets, the timings do not precisely match. Overlain on ASTER GDEM, the Global Digital Elevation Model produced by ASTER, courtesy of NASA/JPL-Caltech.

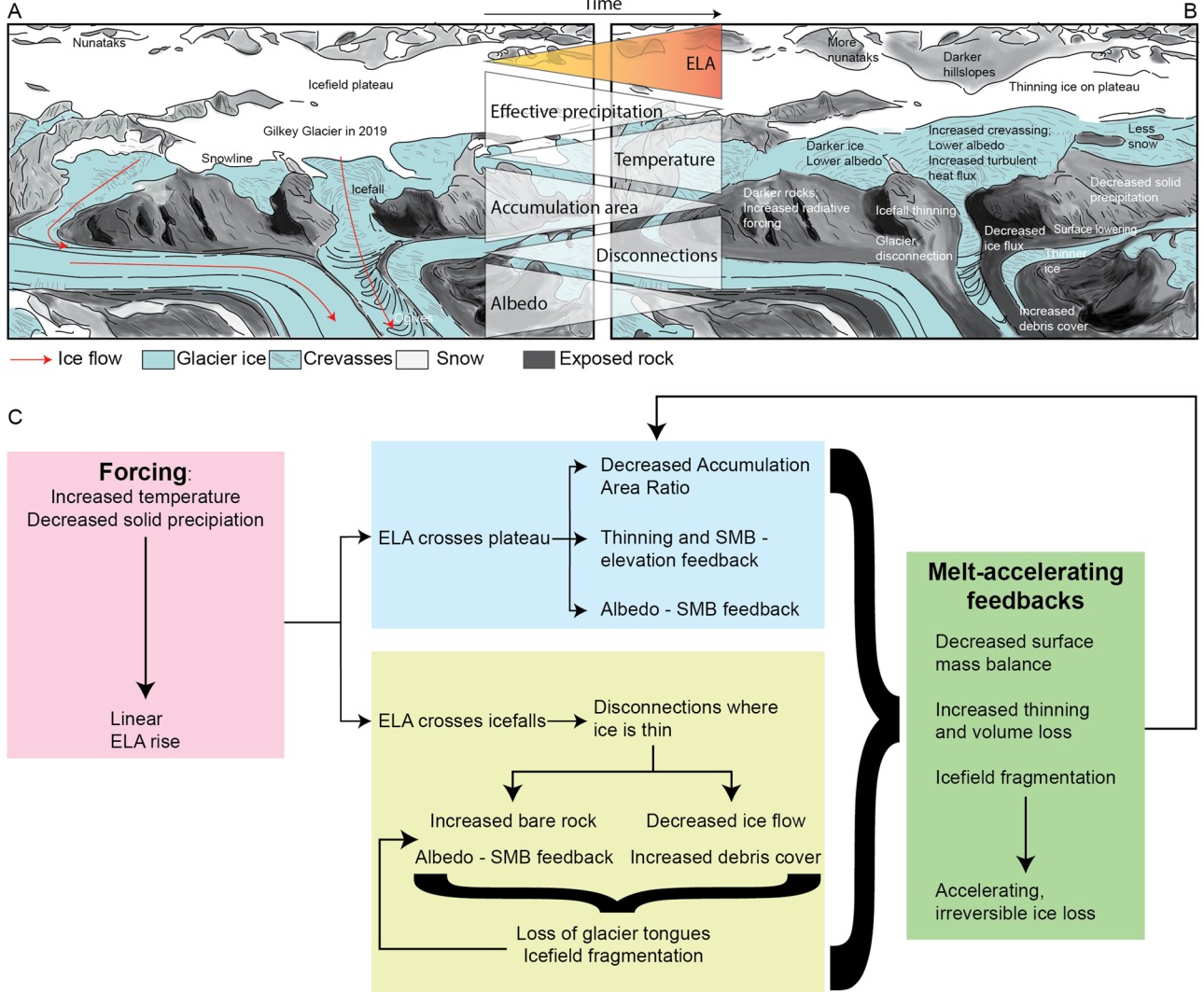

**Fig. 8 | The key processes operating on the low-slope plateau on Juneau Icefield. A** Schematic conceptual sketch of Vaughan Lewis Icefall and tongue on Gilkey Glacier, Juneau Icefield, in 2019. **B** Schematic sketch of same site to illustrate how it might look in a few decades, highlighting some of the key processes. **C** Flow chart illustrating the different processes contributing to the accelerating recession of Juneau Icefield and at other low-slope icefields. See Supplementary Fig. S1D for a field photograph of the Vaughan Lewis Icefall and Gilkey Glacier.

ArcticDEM 'nodata' values were converted to 0. The ArcticDEM holes were then filled with the ASTER GDEM data (filled output upsampled to 2 m cell size to match ArcticDEM) using raster calculator. Comparison of the composite DEM with the original, unmodified GDEM raster yield differences for individual pixels of up to 15 m, which is within GDEM resolution. This likely stems from the resampling carried out as part of the final raster calculation. Hillshade and slope models were derived from the composite DEM mosaic to aid interpretation of glaciological features.

For some glaciers, historical maps and surveys prior to 1948 are available. Here, the maps provided in historical sources (e.g., ref. 28) were scanned, georeferenced, and the termini digitised and included in the available data. As only terminus positions are available, these are provided as polylines at the glacier terminus, annotated with the year (e.g., Fig. 2E, Supplementary Fig. 5).

**Photogrammetry and generation of orthomosaics and DEMs**
Aerial photographs from surveys flown in 1948 and 1979 are available in Earth Explorer (USGS). They are scanned images, without any geospatial data attached. The files are not georeferenced but the database includes an approximate footprint for some images. The surveys were undertaken in July–August 1948 and from a single flight in August 1979, during a period of minimal snow and cloud cover. We selected all available images from the Earth Explorer database (1446 from 1948 AD and 103 from 1979 AD) that intersected a polygon drawn around our study region, along with images that had no footprints in the database but were part of the same flight roll, and that were not damaged, torn or had holes.

For 1979, we used the calibration reports provided by NASA along with the fiducial markers to resample the images to a consistent size. The 1948 images do not have calibration reports but do have fiducial markers, so we used the average position of the fiducial marks to resample the images. We then applied contrast limit adaptive histogram equalization (CLAHE) to ensure the images had consistent brightness. Using MicMac[71], we found tiepoints and calculated the relative camera orientations for each image and processed a relative orthophoto and DEM, using camera coordinates rather than geographic coordinates. The 1948 images were processed in four separate blocks due to the number of images involved. We then automatically registered the resultant orthophotos to a Sentinel-2 image to find ground control points. Elevation was determined using the Copernicus 30 m DEM (https://spacedata.copernicus.eu/collections/copernicus-

digital-elevation-model). From these control points, we extracted the DEMs for each of the four blocks in 1948 and in 1979, using MicMac. For 1948 we co-registered to the Copernicus DEM and cropped the edges of the DEM to deal with edge effects, before stitching the DEMs together, using an established approach[76]. In post-processing, we masked the errors (largely in the accumulation area) and these are given as 'no data'. Voids are filled using the local mean elevation change method[77].

DEMs of difference were then produced for LIA-1948, 1948–1979, 1979–2000 AD. Volume change from 1979–2000 uses the STRM. DEMs for volume change analysis from 2000–2020 are derived from ref. 72, using ASTER imagery and ArcticDEM. DEMs were resampled prior to differencing to the extent of a Copernicus DEM over the study domain, which includes resampling to the 30 m pixel size of the Copernicus DEM. This is undertaken using bilinear resampling. The 1948 and 1979 DEMs are therefore upsampled, as they are higher resolution than the Copernicus DEM.

### Glacier area determination and uncertainty

Initial ice divides and glacier outlines were downloaded from the Randolph Glacier Inventory (RGI) version 6.0[66,67], which has a census date of AD 2005[78]. This historical inventory is our baseline from which recent changes are assessed. The RGI outlines were updated and removed since some were incorrect; they were misclassified snow cover or shadow, or inappropriately subdivided into glaciers. Some glaciers were missing, or were incorrectly delineated, or had over-lapping outlines with another glacier polygon. These errors are clear when comparing outlines to the higher resolution, more recently acquired satellite imagery. RGI 6.0 glacier outlines were therefore manually edited to be consistent with Landsat 7 ETM+ imagery from 2005 (updated outlines published[21], see Supplementary Table 6). Glacier outlines were then manually edited in ArcGIS according to observed changes around the glacier margin from the satellite ima-gery, aerial photograph orthomosaics, 1948 USGS topographic maps, both in the terminus and forefield, and at higher elevations, to reflect glacier extent in 2019, 2015, 2005, 1990, 1979, and 1948 AD. RGI IDs were maintained for each glacier polygon. Where glaciers had sepa-rated into separate polygons, the same RGI ID was used for both and the glacier was treated as a multipart polygon in ESRI ArcGIS. The high pixel resolution of the recent Sentinel-2 images and archival aerial photographs allowed a more detailed glaciological analysis and smal-ler minimum glacier and lake area for mapping than earlier inventories, which relied on coarser-resolution Landsat imagery[78]. The minimum glacier and lake size mappable was 0.001 km². The date for each glacier is noted in the attribute information and is used to calculate annualised rates of change. Glaciers were categorised as glacierets, mountain glaciers, valley glaciers or outlet glaciers, following GLIMS guidelines[79] and previous work[21].

To generate glacier outlines for the latest neoglacial maximum (the LIA), the RGI v6.0 2005[67] outlines were manually edited to extend to the crests of LIA-associated moraines, bare ice-scoured rock and associated trimlines, using published geomorphological mapping[21] and published chronological and geological data[28,32,34,80,81], which are reviewed comprehensively in the Supplementary Information (e.g., Supplementary Fig. 5). The more subtle geomorphological evidence included stripped bedrock at elevations beneath surrounding vege-tated hillslopes, and locally-depressed treelines. Glacier moraines at Herbert, Eagle and Twin Glacier Lake were ground truthed and field-checked during fieldwork in July 2022. Published ages were gleaned from the literature and included in the geodatabase. Radiocarbon ages were recalibrated using CALIB[82]. Calibrated mean ages (2σ) are pre-sented as 'cal. ka BP'. Morphostratigraphic principles[83–85] were applied to differentiate between the most recent, LIA Neoglacial moraines, and older moraines deposited in the Younger Dryas or earlier Holocene glaciations. In most cases, Holocene neoglaciations are likely to have

been within the limits of the LIA[18], which makes confusion of older ice limits less likely. Where it was unclear which moraine related to the LIA, the innermost was chosen, to provide a conservative estimate of LIA area. Cross-checks were then performed against sites that have been documented as dating from the LIA.

Uncertainty in glacier area was calculated following the methods outlined by ref. 86. Eleven representative glaciers, across the spectrum of glacier sizes and including typical examples of debris-covered and clean-ice glaciers, were digitised seven times, with a minimum of one day between each round of digitising (Supplementary Fig. 8). The mean, standard deviation and 95% confidence interval was then cal-culated for each glacier for each timeslice. There was a significant correlation between glacier size and confidence interval (Supplemen-tary Table 7). A regression through the data points provided an equation that was used for size-class specific up-scaling to the full dataset[66,86,87]. As a comparison, we also calculated the standard error and standard error of the forecast for each timeslice. This gave a similar, but smaller, estimate of the uncertainty, and so the confidence interval, the more conservative estimate of uncertainty, was used.

Glacier recession, defined in terms of area change per annum (km² a⁻¹) and proportional rates of area change (% a⁻¹), was calculated for each glacier. Note that where we refer to terminus retreat (i.e., a glacier length change) this is specifically stated. Annual rates of area change were calculated by dividing area change (km²) by the time between analyses for each glacier (time was taken from date of satellite or aerial photograph image acquisition). This allows results to be compared between different times and different glacier areas. For 1948 and 1979, where the same imagery is used for a few glaciers in each timeslice due to a lack of coverage, this means that rates of glacier recession are 0 and are removed from our analysis. However, all glaciers are included in the total summed areas given for each timeslice, and the total summed rates of change. Rates of glacier recession for individual glaciers (% a⁻¹ and km² a⁻¹) are normally distributed. Mean rates are presented with the 95% confidence interval.

### Snowlines

The late summer snowline from 2019 AD was mapped from the cloud-free Sentinel-2 imagery (Supplementary Table 6), using black and white band 4 imagery (in line with structural mapping conducted in ref. 21). These images, from 30th August to 7th September 2019, provided clear, cloud-free coverage of the entire icefield. The short timeframe between image acquisitions allowed temporal continuity and clear visualization of icefield snowlines. Snowlines were identified as a wavy, discontinuous boundary between bright white and duller grey; the higher albedo snow reflecting a brighter white colour.

### Glacier disconnections

Glacier disconnections, where the tongue of a glacier had become separated from its accumulation area[62,63,88], were mapped across the icefield published in ref. 21 without a date stamp. Individual dis-connections were mapped that occurred prior to each timeslice. We classified glacier tongues as 'disconnected' if they were associated with one or more of these disconnections from the accumulation area. In the case of disconnected glaciers, multipart polylines were used to record glacier length along the same flowline. The timing of dis-connection was calculated through comparison with the glacier area outlines, and the slope and elevation of the disconnection calculated through automated tools using tools within the GIS. Elevations were derived from ASTER GDEM[74].

### Surface albedo

We derived a stack of 299 Landsat 5–9 images in Google Earth Engine for the years 1986–2023, limited to WRS path/row 57/19 and 58/19, for all available scenes with less than 20% cloud cover on land (Supple-mentary Table 15). Clouds were masked using a cloud cover filter using

the USGS QA_Mask, removing any pixels identified as cloud by the USGS algorithm. We calculated albedo for each pixel in the stack using the albedo equation developed for Landsat 7[89,90] bands blue, red, and infrared, and applied this across the TM, ETM+ and OLI sensors for Landsats 5–9. We then calculated average albedo for every scene within the icefield boundaries as of 1990, 2005 and 2019 AD (not including the peripheral ice masses). This captured the plateau and glacier tongues. Using the average albedos calculated within the 2019 outlines effectively analyses the darkening of snow and glacier ice only, whereas using the 1990 outlines captures glacier recession and fragmentation in addition to darkening snow and ice. We also calculated the average albedo in each scene for pixels above 1500 m in the Copernicus DEM (~2013 AD), this capturing just the plateau. Observations were limited to at least 60% of the plateau being covered and not masked out by cloud.

### LIA volume reconstruction

Our LIA outlines were converted to lines with points along them at intervals of 100 m, and those points were used to extract elevations from the ALOS DEM[75], which was selected for its seamless coverage. A surface to represent the LIA glacier ice surface below the ELA was interpolated between those points. Carrivick et al.[68,69] have shown that there is a negligible effect of which DEM (resolution and date) is used in these types of analyses, because the points are preferentially located from the high-resolution 2 m Arctic DEM[73] even if the elevation values are extracted from a coarser resolution DEM. However, Carrivick et al.[68,69] have also shown that there is an effect on volume loss calculations of the choice of interpolation method and in this study, we followed their advice using a Natural Neighbour interpolation.

Due to an absence of moraines and trimlines above the equilibrium line altitude of glaciers, our mapping of a LIA glacier outline, our LIA ice surface and our analysis of elevation changes and hence volume loss only pertains to LIA ablation area. Following the method of Carrivick et al.[68,69], LIA glacier ablation areas were produced by automatically estimating glacier-specific equilibrium line altitudes (ELA$_{LIA}$) using the Area-Altitude Balance Ratio (AABR) method[91]. The AABR method was chosen to explicitly consider (spatially-distributed) glacier hypsometry. The glacier-specific ELA$_{LIA}$ were subsequently merged with our mapped moraine crests and trimlines to create a single polygon for each glacier ablation area. We estimated the ELA$_{LIA}$ using a BR of 1.88 for all glaciers as suggested to be typical of maritime glaciers[92]. Published work shows that the choice of balance ratio has little influence on glacier ELA[93].

Volume change estimates presented in this study can only be considered as a proxy for mass balance if it is assumed that glacier surfaces above the ELA have not changed since the LIA. Thus, volume change using the methods in this work should be considered as minimum estimates because probably some mass loss has occurred above the ELA$_{LIA}$ as they have risen in elevation due to atmospheric 20th Century warming. That mass loss above the ELA$_{LIA}$ would likely be via surface lowering, rather than areal contraction, and so we would not detect it with our mapping.

### Glacier volume change calculation

To determine volume changes (LIA-1948, 1948–1979, 1979–2000 AD), we used the LIA, 1948 and 1979 glacier outlines, using the outlines that correspond to the start date of the DEMs of difference for each time period. For each glacier, the outline was divided into elevation bands of either 50 m or, 10% of the glacier elevation range, whichever was smaller. In effect, glaciers with an elevation range of greater than 500 m were divided into 50 m elevation bands. The mean elevation change was calculated for each elevation band, removing outliers that were more than 3 normalised median absolute deviations (NMAD) away from the median of the band. We use the mean value for the elevation band if there is 25% or more coverage by area within the elevation band. If there is less than 25% coverage by area within the elevation band, then we use the icefield-wide average for that elevation band.

For each elevation band, this mean elevation change value was multiplied by the area of the band, to calculate volume change, per elevation band. Each band's volume change was summed, and the sum of this is the total volume change for each glacier. For glaciers with no DEM coverage at all, we use the icefield-wide area-averaged thinning rate to scale up the total volume change for the region. This amounts to 44.6% of glacier area for LIA-1948, 16.8% for 1948–1979, 4.4% for 1979–2000, and 0% for 2000–2020 (Supplementary Table 8).

The sum of the total volume change for each glacier was then summed to calculate total volume change for each time period. Mean dh/dt per glacier was calculated by dividing the mean elevation change value by the time period. Mass change was not calculated. For LIA-1948, where pixels showed a positive elevation change, this was assumed to be erroneous, due to the substantial area change, these pixels were masked out. The LIA-1948 elevation change will be a minimum, because the LIA DEM only covers ablation areas.

We use the normalized median absolute deviation of off-glacier pixels to estimate the uncertainty in the DEMs of difference. To estimate the uncertainty in elevation change ($\sigma_{dz}$) introduced by interpolating elevation bands with no data, we adapt the approach utilised in ref. 33 in Eq. (1):

$$\sigma_{dz} = N_z(1 - f) + (M + N_z)f \tag{1}$$

Where $N_z$ is the NMAD of off-glacier elevation differences calculated between the two DEMs, $f$ is the fraction of the glacier that required interpolation, $M$ is the standard deviation of all on-glacier pixels.

We estimate the total uncertainty in each glacier's volume change as the quadratic sum of the elevation change and the area parts (Eq. 2).

$$\sigma_{dv} = \sqrt{\left(\sigma_{dz}A\right)^2 + \left(\sigma_A \overline{dh}\right)^2} \tag{2}$$

Where $A$ is the glacier area, $\sigma_A$ is the uncertainty in glacier area (following the methods above), and $\overline{dh}$ is the area-averaged elevation change on the glacier.

Uncertainty is higher in the LIA-1948 period due to the interpolation of the datasets over areas with no data. The LIA DEM is calculated only over the icefield ablation areas, and the 1948 DEM is likewise not available over Canada. This does not mean that the change is insignificant, but we emphasise that the uncertainty is large because of interpolation, not because of variability of the elevation differences.

Volume change, and uncertainty, from 2000–2010 and 2010–2020 AD was derived directly from ref. 72. Glacier outlines were taken as the RGI v6.0, as explained in the original publication. Glacier-averaged mean volume change and mean elevation change through time (mean dh/dt) was calculated following the method above. Thus they do not take into account area changes through this time period and should be considered a minimum value.

### Data availability

All data used in this work are available open access and source data are provided with this paper. The glacier outlines, source data, and the novel digital elevation models and orthomosaics from the LIA, 1948 and 1979 are available from Mendeley Data (Davies, Bethan (2024), "Juneau Icefield 1770-2020", Mendeley Data, V3, doi: 10.17632/4djw8z3jrb.3; https://data.mendeley.com/datasets/4djw8z3jrb/2)). Source data are provided with this paper. The updated Juneau Icefield glacier outlines are also available from GLIMS. Shapefiles of geomorphological mapping used in the neoglacial LIA reconstruction are available from Davies et al. 2022 (ref. 21). Historical topographic maps are available from the USGS map viewer (https://ngmdb.usgs.gov/

topoview/viewer/#4/52.19/-123.71) as georeferenced GEOTIFFS. Aerial photographs and Landsat and Sentinel satellite imagery are available from USGS Earth Explorer (https://earthexplorer.usgs.gov/). SRTM available from the USGS EROS archive (https://www.usgs.gov/centers/eros/science/usgs-eros-archive-digital-elevation-shuttle-radar-topography-mission-srtm-1) and the 3" Viewfinder Panorama DEMs are available from https://viewfinderpanoramas.org/dem3.html. ASTER GDEM V3 available from the NASA Jet Propulsion Laboratory (https://asterweb.jpl.nasa.gov/gdem.asp). DEMs of difference from 2000–2010 and 2010 2020 are available from Hugonnet et al 2021 (ref. 72). Source data are provided with this paper.

## Code availability

The software used in this manuscript includes ESRI ArcGIS, QGIS, MicMac and MatPlotLib, as well as standard MS Office programmes. Information on installation, system requirements and instructions for use are all available with these software packages. Information on MicMac is available here: https://github.com/micmacIGN/micmac. Code to derive albedo in Google Earth Engine (GEE account needed): https://code.earthengine.google.com/?scriptPath=users%2Frobertmcnabb%2Fjuneau%3Aalbedo.js

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

## Acknowledgements

We thank numerous colleagues and collaborators for fruitful discussions and support, including Fabien Maussion, Tamsin Edwards, Seth Campbell, and faculty and students from the Juneau Icefield Research Project. J.E. acknowledges support from a Natural Environment Research Council (NERC) independent fellowship (NE/R014574/1). L.N. was supported by research funds from the Faculty of Geo- and Atmospheric Sciences at the University of Innsbruck. Fieldwork to Juneau Icefield by Bethan Davies and Lindsey Nicholson was supported by the Royal Geographical Society and the Geological Society, as well as the Juneau Icefield Research Project. Any use of trade, firm or product names is for descriptive purposes only and does not imply endorsement by the US Government.

## Author contributions

B.D. and R.M. conceived and designed the study. B.D. mapped the glacial geomorphology, snowlines and other glacial features. B.D. and J.B. compiled satellite data and mapped glacier extents. J.B. compiled ERA5 reanalysis datasets. R.M. supported methodological development and processed aerial photography and satellite imagery to derive glacier volume changes, and calculated albedo. J.C. reconstructed ice surface at the LIA. B.D. visualised and analysed the data. C.M. supported methods development and analysed and validated the data. B.D. and L.N. undertook fieldwork to ground-truth geomorphological mapping. M.P. validated ELA measurements and historic ELA observations. T.H. supported data analysis. B.D., L.N., J.C., J.E., B.M. and C.M. supported funding acquisition for fieldwork. B.D. wrote the main manuscript with support from R.M.; B.D., R.M., J.B., J.C., J.E., T.H., B.M., C.M., L.N. and M.P. contributed to scientific discussion and interpretation of the results and all authors contributed to the manuscript.

## Competing interests

The authors declare no competing interests.
