## [Peer Review File · Nature Communications]

Accelerating glacier volume loss on Juneau Icefield driven by hypsometry and melt-accelerating feedbacksReviewer #1 (Remarks to the Author):

Review of the manuscript "Accelerating glacier volume loss on Juneau Icefield driven by non-linear processes" by Bethan Davies et al.

The research led by B. Davies, shows an interesting case study on Juneau Icefield in Alaska about the evolution of the glaciated area and volume of the glaciers in this region. Manuscript authors apply state of the art remote sensing techniques to delimitate glacier extent in different time periods. They are also able to reconstruct glacier surfaces to determine volume differences across distinct temporal windows. The manuscript reads well and shows reliable results for distinct time periods that without any doubt show the fast decline of Juneau Icefield in the last decade compared to previous period with data.

Major comments

Despite I have enjoyed the read of the manuscript I consider that there are some major points to be addressed before considering for publication:

- The manuscript claims that non-linear processes drive the volume loss on Juneau Icefield but this is not really shown in the manuscript. The database demonstrates that volume and area losses have accelerated in the last years and that many glaciers have disconnected or separated. On the contrary processes that control and will control glacier evolution are enumerated and related in a schematic (figure 6). These processes have not been measured or evaluated with the data processed and thus, from my understanding some sections of the manuscript must be changed considering this (including the title). See minor comment for lines 247-248.
- The volume and glacier extent for 1948 and for the LIA maximum, have some uncertainties not included in the analysis. For example the glacier extent in 1948 is based in a 66.6% in images of this year, but the glacier area not observed in this year is inferred from topographic maps, and satellite images from 1970 to 1982. Similarly, LIA extent. Similarly, the uncertainty in area extent at the LIA maximum is of same order of magnitude that the extent in 2005, 2015, 2019, periods for which high spatial resolution satellite observations were available. Despite the LIA extent can be well identified from methods described from lines 250 to 271 of the supplementary information, an evaluation of the uncertainty of this extent must be included. Similarly the ELA reconstruction for the LIA might have an error assessment .
- The readability of some figures must be improved. In some cases small circles avoid the identification of glacier extent in particular time periods. Similarly, some arguments are directly related with figures panels that do not have a correct identification doing difficult to follow the discussion. See minor comments for the figures.

Minor comments:

Line 18-19 and 34: In the second sentence of the abstract it is stated "The majority of the ice volume loss comes from Alaskan icefields,...". I see a bit biased this previous sentence when in line 34-35 it is stated that " with 13 cm coming from glaciers, a 25% of which will come from Alaska". Definitely it is not the majority of volume loss. Change conveniently.

Line 26-27: While it is true that that non-linear processes have to be incorporated into glacier evolution models; this is not a direct conclusion of the work. I would remove this last sentence of the abstract, and show a more general conclusion obtained on the frame of this study.

Line 64: I have found quite long the supplementary information section about LIA ice extent and the climatic conditions observed in this period. I would encourage to reduce the extent or even completely remove these sections and just include appropriate references to this period. Include reference to research articles focusing in this time period and remove the reference to supplementary information.

Line 84: Reconstruction of LIA icefield ablation areas: Provide a short explanation on how was obtained

Line 84: Include uncertainty in the total ice volume estimation for LIA reconstruction " $\pm XX \text{ km}^3$ ".

Same for total cumulative ice loss ($315.31 \text{ km}^2 \pm XX \text{ km}^2$)

Line 110: The extent of 17.54 km^2 was an area decrease, right? It is not clear as in this sentence, it has not been computed as an area loss. In the next sentence it is clearly identified the 3.72 km as a loss. Please, along the manuscript, state clearly when losses are observed, either with a "-" before the

number, or clearly stating that a "decrease" or a "loss" has been observed.

Line 118: M and SD are?? Mean and standard recession rates? Per year? Maybe I have missed the introduction of these acronyms, but I am not able to find them. Please, include them conveniently. Same for "p" and "t"

Line 120: A volume loss of -5.91km³ might be volume gain. Please see comment for line 110 and change conveniently.

Line 126: Rates of glacier shrinkage of area or volume? Please specify.

Line 129: These doubling rates of volume loss after 2010 can be observed in Figure 4? I think they are, please, include which panel in figure 4 show it.

Lines 153-168: The discussion is in some sections difficult to follow as figure 4 panels are not conveniently included.

Line 180: Also include here this reference: <https://doi.org/10.1038/s41558-018-0296-5>

Line 204: The debris-covered areas isolation from the climatic evolution might be very important. I think this can be longer discussed in these lines. Moreover, as glacier continue their area decrease, debris-covered areas will increase thus their impact will be more evident.

Line 224-227: Include references to state where and which are the ice fields in Canada, Greenland, ...

Line 247-248: It is stated that "as our observation demonstrate, once the ELA rises above the plateau non-linear processes operate". However the only reference to the ELA and the non linear processes in the manuscript is this: "The critical driver of this non-linear and accelerating glacier decline at Juneau Icefield is that the rising ELA now intersects the elevations of the icefield plateau (Figure 6)." Figure 6 is an schematic figure, showing processes, but it does not demonstrate with observations that non-linear processes operate.

From my understanding the main message of this manuscript (including the title) cannot be sustained with this figure, and thus must be conveniently changed. Moreover, the ELAs included in the extended data 1, do not include any temporal information. These are recent ELAs?? Please, clarify the ELA's date computation in extended data 1

Online methods:

Line 423: Cloud cover patches are limited in area, but which %?

Line 425-436: The 28% of the glacier extent in 1948 is mapped from the topographic maps from which year?, the 3.7% from 1979 AHAP, the 1.67% from 1982 Landsat... (all data from Supplementary table 2). For me, this is not the true glacier extent in 1948 (more than 30 years in between some acquisitions). This must be clearly stated in the manuscript, and the glacier areas computed for 1948, include a higher uncertainty showing the not negligible % of the study area from a different (and later) observations. A good example where this must be shown is in figure 3 glacier areas.

A good approach to determine if images from 1979, 1981 etc (or the topographic maps) can be exploited to identify glacier limits in 1948, is to analyze in areas with data in 1948 and later years, which is the deviations between these extents.

Line 442 GDEM data is ASTER GDEM?, please clarify.

Line 502: I would only talk about LIA maximum without including a specific year (i.e. 1770) For the maximum LIA extent I do now see relevant + or - 20 years. Same for supplementary table 6.

Line 530: And what did authors do with images from 1980, 1981 and 1982 (supplementary table 2), which are also used for computing 1948 glacier extent?

Line 579-580: From previous sentence this is obvious, remove the sentence: "In effect, glaciers..."

Line 583-584: "Then we" appears twice.

Line 587: Which is the % of glaciers with no DEM coverage?, please include this information here.

Line 590: Mean dhdt should be dh/dt?? Same in line 609

Figures:

Figure 1: I encourage including here a map/satellite image showing the entire study area as this is the first figure of the manuscript and thus readers want to identify the main characteristics of the study site. I would remove panel D, which is not informative at this initial point of the manuscript. From this "complete" overview of Juneau Icefield I would delimitate with a rectangle where panels C and E are located.

Panel A: Include boundaries for a better sea-land limit identification

Panel E. Some contour lines (with transparency) might help to interpret the map. Glacier limits in 1931 and 1908-1910 are informative, but if I am right are not used in the manuscript, and thus are not needed. Disconnection separation marks are barely identified in panel E. I encourage to include here not only circles with distinct colors and use triangles, squares etc...for easier identification.

Figure 2: The number of polygons (in brackets), is not informative of the number of glaciers, I would remove this information and change it by the total extent of the ice-dammed lakes, alluvial fan,... ELA (LIA) circles hamper the identification of the estimated LIA glacier extent. For me is more informative the LIA glacier extent (dark right) than the ELA estimation. I would remove all circles in A and include a "C pannel"(or include a new figure in the supplementary material), just including ELA (LIA) with features of different colors (with same color legend as these of the small circles).

Figure 3: Again, the number of polygons in brackets is not informative. Include area extent instead the number of polygons. Regarding the volume rate of change, the error must be also included. Disconnection and separation circles are barely seen in the figures. I encourage manuscript authors to change symbols and colors for a proper identification.

Figure 4: Panel letters are not included (a), b), c)... what makes difficult figure interpretation. Figure 4 caption: Summer air temperatures are computed for which time period?, please, include it.

For panel b) is stated "The timing of an early, mid and late LIA is shown." Where is this shown? I think this is an interpretation of the graph, at least nothing is marked in the graphs. If this sentence is an interpretation I would include it in the text not in the figure caption.

For the bottom left panel, please include a legend for the red line and the green triangle (mean and median I guess).

Juneau airport data are used in the manuscript but it is not located in any figure or the distance to the study area is included. Please, provide, at least the distance from the airport to the ice field and the elevation of the weather station in the text/caption of this figure.

Extended data 1: Please include the dates for which the ELA was computed. Hypsometric indexes are used in the manuscript? I guess these values are not used. Please, if so, remove them in this figure.

Supplementary information:

Line 57 and 60. Please include precipitation in mm (International system units)

Line 76 There is a typo in the year. I guess it is not 2948.

Line 156 : Where is Taku Point?? Show it or include a refence here.

Supplementary figure 1: 0-100m and 100-200 m, ice thicknesses representation can be improved with more classes (0-50, 50-100....). This would show the areas with extreme glacier retreat in the forthcoming years.

Supplementary Figure 3: Include the location of the study area in NW North America (at least a point with the location or an small rectangle).

Supplementary Figure 5: I might remove the 5 upper glaciers in this figure as it is difficult to identify the distinct lines. From my understanding is better to show the methodology in 2-3 glaciers with good examples than showing it with 10 examples barely observed.

Reviewer #2 (Remarks to the Author):

This paper reconstructs ice volume and areal changes for the Juneau Icefield (JIF), a plateau icefield with outlet glaciers, and adjacent glaciers from the 19th century to 2019, using multiple data sets, including geomorphological and topographic maps, air photos and satellite imagery. The most noteworthy finding is that the rate of ice loss from the JIF has not been linear and has accelerated sharply since 2005 and that the acceleration is due, not to an acceleration in temperature increases, but is due to the JIF hypsometry, as the ELA for the icefield has now risen to the plateau surface. This means that ablation zone is now rapidly expanding over a wide area and the rate of ice loss will continue to increase even if temperature increases stay low.

These findings are of significance to glaciologists and climate and sea-level modellers for a number of

reasons. Most importantly, the study suggests that current model projections of ice loss and resulting sea level rise from Alaskan and other ice fields may be underestimates, as they are based on a linear rate of ice loss. This has important implications for the rate of sea-level change in the next 100 years, as Alaskan glaciers are an important contributor in the short term.

Secondly, the paper provides a model for the mechanisms of ice field retreat which is applicable to both modern and past ice fields globally. The paper builds on previous studies of Juneau ice field and is a significant new contribution to the literature. The use of sequential DEMs to assess ice loss is not new, (see, for example Berthier et al., 2010 and 2018), but this approach has been previously applied to changes in this century only. The generation of a sequential DEMs covering the period from 1770-2020 is an important achievement, and is a result of combining multiple datasets, including geomorphological and topographical maps, air photos and satellite imagery, to produce the time series.

The methodology used here is sound and the resulting data is of high quality; possible errors are discussed in detail in the methods and supplementary data expands on some of these data sets. There is sufficient detail provided to reproduce the work. However, I do have some issues with the conclusions drawn from the work presented. This is due more to the writing style and presentation of data within the main text than to any inaccuracies in approach. Nevertheless, some rearrangement/minor rewriting is necessary to highlight particular aspects of the data and make it clear when the authors are referring to their own or other datasets. In particular, I think the following needs addressing:

1. A key finding of the paper is that the rising ELA of the Juneau Icefield now intersects the elevations of the icefield plateau; this is stated without preamble on line 171. However, the only clear ELA data presented in the results section is for the LIA. Instead, evidence is presented for thinning across the icefield between 2010-2020, plus a large increase in disconnections in this period. However, this data is buried within a paragraph (lines 114-124) and it took me some time to figure out that this must be the supporting evidence for the statement on line 171, and that this is clearly different to previous periods. The authors should clarify if this is the case. If it is not, then they need to make clear what the evidence is for the ELA rising to the plateau height.
2. Line 178 – It is not clear here whether the dramatic reduction in icefield accumulation area refers to your data or the predicted changes. I assumed on first reading it was the predicted change, as the authors have not calculated a change in accumulation area volume, so cannot say it is dramatic as yet? If this does refer to observations presented in the paper, the authors need to make it clear why specifically they think the accumulation area has decreased dramatically, as this would require retreat of the ELA beyond the edge and toward the interior of the plateau.
3. Lines 178- 196, I think, refer to the model of ice recession presented in Fig. 6, but the absence of an initial linking statement to this effect means that it reads as if the work presented here directly measure the changes implied, e.g. snow cover duration. This fails to distinguish between the authors' own work and that of others adequately and needs addressing.
4. I was surprised that there is no reference to the two papers by Berthier et al. (2010 and 2018) on estimating recent rates of change for Alaskan including JIF (2010) and JIF specifically (2018). The 2018 paper is mentioned in passing, but both papers show an earlier attempt to estimate ice loss using DEMs, although at a lower resolution, with much less coverage and for a much shorter time. In particular, it is worth considering a comparison of the authors' work to the 2018 Dh/Dt work on specific glaciers, to see if there is a significant change. This need not be included in the main paper, but should be in the extended methods?

There are other minor issues/typos within the figure and text to address:

Line 58: "way a.." – remove a

Figure 1: I didn't find this particularly helpful, other than the map showing the location of the JIF. Consider merging 1A with Figure 3 (but see below re Figs 2-3) and moving the rest to the supplementary/extended sections.

Line 83: I would extend this to say why Lemon Creek is used specifically – presumably because of the excellent data set? But not everyone knows that as a matter of course.

Line 88 – worth saying "underwent dramatic changes *in area* from the LIA.." – I had to reread the paragraph to figure that out. I recommend checking throughout that you have been precise in

specifying area/volume etc. – it's quite confusing otherwise.

Figure 2A: the ELA circles almost entirely obscure the smaller glaciers so it is hard to see the LIA extent vs the 2019 extent. You could make the circles smaller, but a better alternative might be to change the order of the figures – put Figure 3 before Figure 2 as it shows the areal extent clearly. There is also room to add a v small map showing the location of the JIF on Fig. 3.

Figure 2b: The figure caption vs the legend is confusing. The legend implies all the outlet glaciers except for Taku have increased in height, but I think the inverse applies? i.e. the figure caption is correct and this shows the amount of lowering, rather than the amount of change.

Line 100 – also refer to your Figure 3.

Line 107: grammar

Line 115: grammar – should be "...areal loss accelerated sharply..."

Line 121: you need to distinguish more clearly between the "icefield plateau" (above) and the icefield (this line) – suggest "the whole icefield". I also got confused in places between glaciers (presumably including outlets) and glaciers (not including outlets possibly?) – maybe worth saying "all glaciers" when you mean the lot.

Lines 119-122: key points buried in the middle of a paragraph – need to be highlighted....

Line 125: this reads as a contradictory statement – i.e. "rates of areal recession rise steadily from 1948-2005" compared to lines 107 and 128, where you mention the increase in rate from 1979-2010. Needs clarifying.

Line 130: - do you mean ice loss? Or have glaciers completely disappeared? – if the latter, more info needed!

Figure 3: Love this, but the disconnection and separation dots are a little hard to see – brighter colours needed?

Figure 4: It was a pain relating the captions to the graph. I suggest you rearrange the caption and lettering, going across instead of down, i.e.

a b

c d

e f etc.

Line 155: I strongly recommend making your overall point at the end of the first sentence, i.e. temperatures important, but the relationship between temperature and volume/area loss is non-linear. Otherwise this is a very disjointed read. You can still make it at the end and suggest a threshold response.

Line 171: see my point above – you haven't drawn the links back to the data.

Line 176: add "... observational data *presented here*..." – all very abstracted otherwise.

Line 178: see main comments above

Figure 6: This is ok, but the photograph caption is not accurate – that's not the JIF, just a small area of it.

Line 204: refer to supplementary evidence and reference 17.

Line 234: grammar...

Line 274: I think it would be better if 17 was not in superscript – it forms part of the main sentence, it's not the equivalent of an asterix. The same replies to references 46 and 17 below, and references in the extended data where the reference number is part of the main sentence.

Line 466: you mention that 1979 was in 4 blocks below – include here with 1948?

Lines 587: how many are there?

Line 590: This could be compared to the Berthier et al. (2018) example to see if your DEMs make a difference in calculating area/volume?

Reviewer #3 (Remarks to the Author):

The authors present a compilation of observations and reconstructions of changes of the Juneau

Icefield that is probably unique for a glacierized mountain chain of this scale in its spatial and temporal scope, and in its comprehensiveness. Similar densities of observations are found, if at all, in regions like the Alps or Scandinavia, which arguably are less relevant than Alaska to many glaciological research questions. These other data sets have not been created because of their urgency, but because of convenience and traditions. The authors therefore deserve a lot of praise for their effort to improve the knowledge in a region that is much more relevant and much harder to investigate, and they have obviously succeeded in connecting the different qualities, quantities and characteristics of their data to build a panoramic view of the evolution of the Juneau Icefield since the Little Ice Age.

There are therefore very strong arguments that this work should be published; it would be of high value to the glaciological research community (and probably beyond). However, I see two fundamental problems in the manuscript that are not related to the data, but to their interpretation and the framing of the entire manuscript.

(1) The smaller of these problems concerns the discussion of the relevance of the findings with respect to glacier models and projections of glacier evolution. The manuscript claims that "Current analyses of future glacier behaviour predict a linear future contribution of global (8) and Alaskan ice masses to sea level rise (5,6,8), and a linear rate of volume loss for Juneau Icefield (9).", and that "the simplified physics deployed by many of the models in Glacier MIP [...] accounts poorly for variation across different hypsometries (11,14)". The manuscript argues that the nonlinearities supposedly found in the data (see below for more on this) cannot be captured by the current generation of models. I strongly disagree with this assessment. The given references (5, 6, 8, 9) show nonlinearities in the mass change rates over the 21st century, such that invoking "such linear projections" does not adequately reflect their characteristics. It is true that regional and global scale projections "hide" nonlinear behavior of individual glaciers in the aggregated numbers typically shown. This does not mean, however, that this kind of behavior is not represented at the scale of individual glaciers (and it even shows in the aggregate, as stated above). In particular, the well-known feedbacks between glacier hypsometry/geometry and mass balance referred to in the manuscript are well represented in state-of-the-art global glacier models as used in references 6, 8, and 9. Some other nonlinearities are possible in some, but not all of the global-scale models (e.g., concerning disconnections). There are additional nonlinearities which are not captured well by any of this type of model (the manuscript mentions some), these are addressed, e.g., in ref. 14. But those nonlinearities are not the ones the manuscript claims to be affecting the Juneau Icefield.

This issue does not affect the center of the manuscript, and because of this, I see this as the "smaller" of the two fundamental problems. I see it as fundamental since it affects a main motivation of the manuscript, being prominently given in the abstract (and repeatedly mentioned in the manuscript). It should be possible to reframe the manuscript so that it is more adequately referring to the state-of-the-art in glacier modeling.

(2) The bigger of the two problems concerns the claim that nonlinear processes are driving the mass loss acceleration of the Juneau Icefield (see title).

The manuscript has not convinced me that there is in fact a nonlinear response of the icefield observable in the data. Essentially, this conclusion would have to rest on the scatter plot in Fig. 4j. Here, a nonlinear response of the rates of area and thickness change to temperature change would have to be visible. With only 4 (thickness) and 6 (area) point available, and quite a bit of scatter (which might be explained by precipitation anomalies), the argument for nonlinearity is weak. The statement in lines 165 – 168 is therefore not justified; neither the part concerning the present manuscript, nor concerning the modeling studies (see above). Arguing the other way around ("nonlinear responses found in projections are not yet robustly detected in observations/reconstructions") would be a similar misstatement, but arguably not worse than what is said in the manuscript. As an alternative to showing a nonlinear response of the entire icefield (Fig. 4j), it would be possible to demonstrate that nonlinear effects are affecting the glaciers by showing

that the ELA has crossed into the plateau region (i.e., by looking at ELA changes and how they relate to the hypsometry of the glacier), or by investigating the mass balance profile and ice flow velocities in the region of glacier disconnections (see explanation of feedbacks around Fig. 6). Unfortunately, this is not done in the manuscript.

In this context, the section "Hypsometric controls of icefield behavior" is very problematic: the relevant mechanisms are well-known, but they are also explained (and referenced) well in the manuscript. However, the manuscript does not show that a single one of them is actually affecting the Juneau Icefield in their data. Remarkably, they use an edited photograph (Fig. 6) for illustrating "how it might look in a few decades". That is exactly the point: in this manuscript, I don't find any evidence that these feedbacks are active in present day. However, the manuscript very boldly claims that it is the case in the title, the abstract, scattered in the text, and in the final, conclusive paragraph.

The problem of a missing demonstration of nonlinearity in the data may arise from a confusion of "nonlinearity" and "acceleration". E.g., an acceleration in mass change can be a linear response to an increased temperature anomaly. The feedbacks explained in the second and third paragraph of the manuscript (and Fig. 6) would lead to an acceleration in mass change without additional changes in the temperature anomaly. Therefore, I would suggest that the authors carefully decide whether they actually want to claim (and then have to demonstrate) a nonlinear response of the Juneau Icefield to climate change (which, with the data on hand, seems to be very difficult), or whether they wish to make a case for an impressive acceleration of glacier change in Alaska in the recent decades, which would be a very strong case.

Either way, very substantial work on the text (and potentially, the data analysis) would be needed, so that I would probably consider the manuscript as "new". This would imply a suggestion to reject the manuscript. However, as I am convinced that the work definitely should be published eventually, I would also agree with considering this as a case of "major revisions". Expecting the manuscript to go through another round of revisions (potentially in a different format, pending the editor's decision), I have refrained from specific/minor comments on all parts of the manuscript except the main text, where I hope also these specific comments will be helpful (also, I don't see any major issues in the methods).

Specific/minor suggestions for the main text:

L18: „The majority..." would imply > 50 %, which is not the case according to Hugonnet et al. (<https://doi.org/10.1038/s41586-021-03436-z>); perhaps change to: "The largest regional contribution..."

L19: "...projected to become even more dominant..." I'm not sure I would agree, and would expect that this depends on the scenario: in a high-emission scenario, e.g., I would expect that regions that have more ice left (peripheries of Greenland and Antarctica, the Canadian Arctic) become more dominant over time.

L22: "...with..." since this refers to the acceleration (but no acceleration value is given), perhaps change to: "...leading to..."

L35: suggest to change to: "Under the higher-emission scenario RCP 4.5..."

Fig. 1 is a bit disorganized: e.g., I can't find any ice-dammed lakes, lakes, or sandurs on the map (or photographs), and referring to north- and south-flowing glaciers in panel d, without including a north arrow, is not ideal (perhaps label the glaciers in panel d directly?). I would also expect that the individual glaciers mentioned in the text (e.g., lines 83-87) appear in the map, or vice versa. Please include a reference to (e.g.) Fig. 2a in the relevant sections of the text.

L82-90: please indicate the uncertainties of the assessed changes in ELA, volume, and thickness. These uncertainties are important for understanding the significance of the changes.

Fig. 2: I appreciate the difficulty to put the rich data set on the map(s), and unfortunately, I don't have a good solution. But it is problematic that (i) the circles of the ELA estimates cover a large fraction of the trimline and moraine markers and (ii) moraine polygons are shown gray on gray background.

L118: what do "M" and "SD" mean? What units are used here?

L129: It has not been shown that the acceleration is really exponential; I would suggest to state this more carefully (also in the abstract and other places in the manuscript).

Fig. 3: The tables in the lower left should either really be formatted as tables (outside of the figure) or left out (they are already shown in Fig. 4). Regarding number of glaciers: how are separations treated? Do they increase the number of glaciers?

Fig. 4a: please explain red line here, not just in panel f. b: Where is the timing of an early, mid and late LIA shown? Last sentence of the caption: what does this refer to? Panel f does not show reduced precipitation during 2005 - 2015?

Fig. 5: legend and caption: please correct from dhdt to dh/dt. More generally, I would suggest to move this figure to the supplement; it is not that central to the arguments in the manuscript.

Reviewer #1 (Remarks to the Author):

Review of the manuscript “Accelerating glacier volume loss on Juneau Icefield driven by non-linear processes” by Bethan Davies et al.

The research led by B. Davies, shows an interesting case study on Juneau Icefield in Alaska about the evolution of the glaciated area and volume of the glaciers in this region. Manuscript authors apply state of the art remote sensing techniques to delimitate glacier extent in different time periods. They are also able to reconstruct glacier surfaces to determine volume differences across distinct temporal windows. The manuscript reads well and shows reliable results for distinct time periods that without any doubt show the fast decline of Juneau Icefield in the last decade compared to previous period with data.

We are delighted that the reviewer found the manuscript enjoyable and reliable. Thank you for the constructive and helpful comments.

Major comments

Despite I have enjoyed the read of the manuscript I consider that there are some major points to be addressed before considering for publication:

- The manuscript claims that non-linear processes drive the volume loss on Juneau Icefield but this is not really shown in the manuscript. The database demonstrates that volume and area losses have accelerated in the last years and that many glaciers have disconnected or separated. On the contrary processes that control and will control glacier evolution are enumerated and related in a schematic (figure 6). These processes have not been measured or evaluated with the data processed and thus, from my understanding some sections of the manuscript must be changed considering this (including the title). See minor comment for lines 247-248.

There is a substantial data (McNeil et al., 2020; McNeil et al., 2019; O'Neel et al., 2019; Pelto, 2019) that shows the rising ELAs and transient snowlines in Juneau Icefield, which now reach the plateau. This was rather buried in the Supplementary Information, but has now been moved to the main text. We have emphasised this on figure 6i by showing the height of the plateau with respect to the height of the ELAs,

Secondly, we now include an analysis of the late summer snowline on Juneau Icefield in 2019 AD from sentinel 2 imagery. This late summer snowline clearly intersects the plateau and demonstrates the significance of the icefield hypsometry in sensitivity to rising snowlines. We have noted this in the Results section:

“This rapid glacier recession was accompanied by ELAs rising to the plateau height in the period 2005-2020 (McNeil et al., 2020; McNeil et al., 2019) (Figure 6i), and exceptionally high snowlines on the plateau in 2018 and 2019 (Pelto, 2019) (Figure 5). Late summer snowlines mapped in 2019 from Sentinel 2 imagery (Figure 5) had a mean altitude of 1600 m asl (SD 161 m; n=163). Snowlines on Taku Glacier were a mean altitude of 1444.7 m in 2019; this compares with a measured ELA in 2019 of 1528 m (McNeil et al., 2019). The snowline here is therefore well above the southern plateau at 1200 m asl. No snow was observed on Lemon Creek Glacier, in line with the measured ELA of 2023 m that year (McNeil et al., 2016). Snowlines on Meade, Llewellyn and Field Glacier had a mean height of 1671 m (SD 119 m), well above the height of the northern plateau at 1500 m asl (Figure 5).”

This rising ELA then results in a number of processes related to albedo, thinning on the plateau, shrinking accumulation area, disconnections and so on, which are enumerated in the Figure 6 (now Figure 8). We have edited the Discussion, highlighting the rise in the ELA and its intersection with the icefield plateau.

Finally, in line with suggestions from Reviewer #3, we have reframed the abstract and discussion on conclusions around the accelerated mass change observed driven by the ELA and glacier thinning now intersecting with the icefield plateau, which drives a number of hypsometrically driven melt-accelerating

feedbacks (e.g. related to albedo, thinning, accumulation area ration, fragmentation). Therefore we have strengthened the discussion with reference to these processes and hope that this is now addressed.

- The volume and glacier extent for 1948 and for the LIA maximum, have some uncertainties not included in the analysis. For example the glacier extent in 1948 is based in a 66.6% in images of this year, but the glacier area not observed in this year is inferred from topographic maps, and satellite images from 1970 to 1982. Similarly, LIA extent.

We have clarified the data sources and analysis in the text and in the response to reviewers below. All uncertainties are carefully included in the manuscript. For 1948 in particular, careful analysis of the topographic maps used is included in the supplementary information, including the year used for the imagery. This date (of either the year of publication or the year if the imagery used) is included in the attribute information for this dataset – all of which is now available in GLIMS viewer (<https://www.glims.org/maps/glims>).

The date of the imagery used is used to calculate annual rates of recession. For those small number of glaciers (12.5% of glaciers) where the earliest available date is the 1980s Landsat imagery, this is included in the attribute information as well. Since glacier shrinkage has occurred consistently across the icefield over this time period, this means that the 1948 glacier area is a minimum. This is detailed in the Methods and in the Supplementary Information.

Similarly, the uncertainty in area extent at the LIA maximum is of same order of magnitude that the extent in 2005, 2015, 2019, periods for which high spatial resolution satellite observations were available. Despite the LIA extent can be well identified from methods described from lines 250 to 271 of the supplementary information, an evaluation of the uncertainty of this extent must be included.

We have conducted a thorough analysis of uncertainty in glacier area extent for each timeslice (supplementary table 6). The moraines, trimlines and areas of ice scour are easily identified and these data have been published in detail already (Davies et al., 2022). The use of Sentinel satellite imagery for these geomorphic landforms, as opposed to the lower resolution Landsat imagery available in the 1990, 2005 and 2015 timeslices, allows a high degree of confidence in the mapped geomorphology. These geomorphic details are briefly summarised in Supplementary figure 4.

The detailed geomorphological mapping allows glacier outlines at the LIA to be drawn with precision and the detailed assessment of uncertainty in Supplementary figure 6 follows best practice guidelines (Paul et al., 2017). There is limited scope for greatly differentiated LIA glacier extent because glacier tongues largely remain constrained in topographic valleys, with a very clear geomorphic imprint (and with few marine terminating glaciers). Therefore the uncertainty in glacier extent as presented in the supplementary information at the LIA is robust.

Similarly the ELA reconstruction for the LIA might have an error assessment.

Uncertainty in ELA reconstruction for palaeoglaciers derives from choice of balance ratio (Rea, 2009). Analysis by our team members on the influence of different balance ratios on glacier ELA shows that ELA is only weakly sensitive to choice of balance ratio (Carrivick et al., 2023). Added to manuscript:

“Published work shows that the choice of balance ratio has little influence on glacier ELA (Carrivick et al., 2023).”

In detail, Carrivick et al. (2023) their supplementary information Figure SI_8 shows that varying BRs between 1.2 and 2.6 [noting Oien et al., (2022) recommend 1.56 for paleoglaciers and Rea et al., suggest 2.24 for arctic glaciers] produces mean ELAs varying between 1040 and 950 m. asl.

- The readability of some figures must be improved. In some cases small circles avoid the identification of glacier extent in particular time periods. Similarly, some arguments are directly related with figures panels that do not have a correct identification doing difficult to follow the discussion. See minor comments for the figures.

Figures have been improved as suggested. Panels are all labelled.

Minor comments:

We have made all of these amendments as suggested, which has improved the clarity of the manuscript. Thank you for these suggestions.

Line 18-19 and 34: In the second sentence of the abstract it is stated "The majority of the ice volume loss comes from Alaskan icefields,...". I see a bit biased this previous sentence when in line 34-35 it is stated that "with 13 cm coming from glaciers, a 25% of which will come from Alaska". Definitely it is not the majority of volume loss. Change conveniently.

Amended: ". The glaciers experiencing the most ice loss are the Alaskan icefields"

Line 26-27: While it is true that that non-linear processes have to be incorporated into glacier evolution models; this is not a direct conclusion of the work. I would remove this last sentence of the abstract, and show a more general conclusion obtained on the frame of this study.

This is the important implication and wider significance of the work. It is one of our key findings and is supported by our data, as well as published mass balance observations across the icefield. Therefore, it should be included as it is the key finding. We have moved the glaciological and mass balance observations from the Supplementary Information to the main text, which now supports this argument more clearly.

Line 64: I have found quite long the supplementary information section about LIA ice extent and the climatic conditions observed in this period. I would encourage to reduce the extent or even completely remove these sections and just include appropriate references to this period. Include reference to research articles focusing in this time period and remove the reference to supplementary information.

We argue that the climate and evidence for glacier extent in the LIA is a crucial part of the paper, and a comprehensive recent review of this is not available in the literature. This data is needed as it explains why the glaciers have been dated to a particular timeframe. There is some complexity in the behaviour of some glaciers, which requires detailed explanation. However, we have shortened and condensed this section of the Supplementary Information, and removed unnecessary detail.

Line 84: Reconstruction of LIA icefield ablation areas: Provide a short explanation on how was obtained.

This is the area below the reconstructed ELA and is explained in line 559 in the Online Methods ("Reconstruction of ELAs and ablation areas).

Line 84: Include uncertainty in the total ice volume estimation for LIA reconstruction " $\pm XX \text{ km}^3$ ". Same for total cumulative ice loss ($315.31 \text{ km}^2 \pm XX \text{ km}^2$).

Amended. The full details of uncertainties are additionally provided in extended data 4.

Line 110: The extent of 17.54 km^2 was an area decrease, right? It is not clear as in this sentence, it has not been computed as an area loss. In the next sentence it is clearly identified the 3.72 km as a loss. Please, along the manuscript, state clearly when losses are observed, either with a "-" before the number, or clearly stating that a "decrease" or a "loss" has been observed.

We try to be clear in the manuscript by being consistent in use of signs. Talking about glacier shrinkage and then giving a "-" sign can be confusing and a double negative. This sentence is talking about rates of glacier

area loss. We have been clear throughout the manuscript and careful to note that these are rates of glacier area or volume loss.

Line 118: M and SD are?? Mean and standard recession rates? Per year? Maybe I have missed the introduction of these acronyms, but I am not able to find them. Please, include them conveniently. Same for “p” and “t”

M is Mean, SD is Standard Deviation. They are commonly used terms. Defined now in manuscript:

“Rates of recession were 1.5 times faster from 2015-2019 (Mean (M) =5.88, Standard Deviation (SD)=5.26)”

t is the T statistic, p is the p value. These are given in the usual APA statistics format.

Line 120: A volume loss of -5.91km³ might be volume gain. Please see comment for line 110 and change conveniently. Amended as asked

Line 126: Rates of glacier shrinkage of area or volume? Please specify. Amended.

Line 129: These doubling rates of volume loss after 2010 can be observed in Figure 4? I think they are, please, include which panel in figure 4 show it. Amended.

Lines153-168: The discussion is in some sections difficult to follow as figure 4 panels are not conveniently included. Amended as asked

Line 180: Also include here this reference: <https://doi.org/10.1038/s41558-018-0296-5>. Amended as asked.

Line 204: The debris-covered areas isolation from the climatic evolution might be very important. I think this can be longer discussed in these lines. Moreover, as glacier continue their area decrease, debris-covered areas will increase thus their impact will be more evident. Noted in the manuscript.

Line 224-227: Include references to state where and which are the ice fields in Canada, Greenland, ...

There isn't a single reference that compiles these. These are icefields named in maps largely. We have added references where available or relevant.

Line 247-248: It is stated that “as our observation demonstrate, once the ELA rises above the plateau non-linear processes operate”. However the only reference to the ELA and the non linear processes in the manuscript is this: “The critical driver of this non-linear and accelerating glacier decline at Juneau Icefield is that the rising ELA now intersects the elevations of the icefield plateau (Figure 6).” Figure 6 is an schematic figure, showing processes, but it does not demonstrate with observations that non-linear processes operate.

Amended to “once ELAs rise and thinning occurs on the plateau”. However, we have ELA reconstructions from published work (McNeil et al., 2020; O'Neel et al., 2019; Pelto, 2019). These ELAs are shown in figure 4i, and the height of the plateau (seen in figure 6) is now described in the main text. The ELAs and glaciology were previously outlined in detail in the SI. We have moved this information to the main manuscript, which aids clarity considerably, as it is much clearer that the rising ELAs are driving the thinning observed on the plateau. Secondly, we have added data on transient snowlines from 2019, following on from published work (McNeil et al., 2020; Pelto, 2019), showing that the snowlines are reaching the plateau now.

The schematic Figure 6 (now Figure 8 due to restructuring) illustrates the processes that are driving the behaviour observed on Juneau Icefield, and is supported by reference to other published work and observation of the rising ELAs.

From my understanding the main message of this manuscript (including the title) cannot be sustained with this figure, and thus must be conveniently changed.

We have updated the main message of the manuscript and changed the title in line with this.

Moreover, the ELAs included in the extended data 1, do not include any temporal information. These are recent ELAs?? Please, clarify the ELA's date computation in extended data 1.

This figure (now Figure 1) provides the most recent available mean ELAs – references are provided in the figure caption. The dates for the ELAs are added to the figure.

Online methods:

Line 423: Cloud cover patches are limited in area, but which %?

Overlapping imagery is used to ensure that cloud cover does not impede mapping. Images listed in the supplementary information.

Line 425-436: The 28% of the glacier extent in 1948 is mapped from the topographic maps from which year?, the 3.7% from 1979 AHAP, the 1.67% from 1982 Landsat... (all data from Supplementary table 2). For me, this is not the true glacier extent in 1948 (more than 30 years in between some acquisitions). This must be clearly stated in the manuscript, and the glacier areas computed for 1948, include a higher uncertainty showing the not negligible % of the study area from a different (and later) observations. A good example where this must be shown is in figure 3 glacier areas.

This is shown clearly in Supplementary Figure 6. As is shown in this figure, the 1948 aerial photo mosaic covers most of the glacier termini in the west, north and south of the icefield. With the exception of Llewellyn Glacier, the icefield outlet glaciers are well covered, and it is the small peripheral glaciers to the east of the icefield that mostly lack coverage from aerial photos. The outlet glaciers not covered by the photo mosaic are covered by the topographic maps. These smaller glaciers mapped from Landsat imagery account for only a very small proportion of glacier area, and therefore uncertainties here are unlikely to affect the conclusions of the paper. For the photo mosaic, larger patches not covered remain over the accumulation area. Where there is coverage there is little difference observed in the accumulation area, and uncertainties or differences here are unlikely to affect the conclusions of the manuscript.

A good approach to determine if images from 1979, 1981 etc (or the topographic maps) can be exploited to identify glacier limits in 1948, is to analyze in areas with data in 1948 and later years, which is the deviations between these extents.

Firstly, we note in the manuscript that the date of image acquisition is provided in the attribute information for each glacier, and the rate of recession provided is annualised using this date.

Secondly, we note in the manuscript that the area for 1948 is a minimum, as a small amount of glacier area is calculated using later imagery. Therefore the area in 1948 is the best available estimate, and is clearly acknowledged to be a minimum. We have added text to the Methods emphasising this.

Thirdly, for the 117 glaciers mapped from topographic maps (supplementary table 3), detail on the date of the topographic maps is provided in Supplementary Table 5; this includes the date of the map published and the number of glaciers used. The published date of the map is used as the date for the glacier attribute information if clear information on source data is not provided.

Added to the Methods in the manuscript:

“This means that the glacier extent in 1948 is the best available estimate given the data availability. Annual rates of recession are calculated using the date of the imagery or data source used to map glacier extent. The glacier extent in 1948 is considered a minimum; use of later datasets results in some glaciers being underestimated due to continued glacier recession.”

Line 442 GDEM data is ASTER GDEM?, please clarify. **Yes, amended.**

Line 502: I would only talk about LIA maximum without including a specific year (i.e. 1770) For the maximum LIA extent I do now see relevant + or – 20 years. Same for supplementary table 6. **Amended**

Line 530: And what did authors do with images from 1980, 1981 and 1982 (supplementary table 2), which are also used for computing 1948 glacier extent?

As stated, if the same imagery was used, then the glaciers are removed from analysis of rates of recession.

Line 579-580: From previous sentence this is obvious, remove the sentence: “In effect, glaciers...”

Yes, this may be obvious, but we retain for clarity.

Line 583-584: “Then we” appears twice. **Amended**

Line 587: Which is the % of glaciers with no DEM coverage?, please include this information here.

This data has been added to the Supplementary Information (Supplementary Table 8). Added to the text:

“This amounts to 44.6% of glacier area for LIA-1948, 16.8% for 1948-1979, 4.4% for 1979-2000, and 0% for 2000-2020 (Supplementary table 8).”

Line 590: Mean dhdt should be dh/dt?? Same in line 609 **Amended throughout**

Figures:

Figure 1: I encourage including here a map/satellite image showing the entire study area as this is the first figure of the manuscript and thus readers want to identify the main characteristics of the study site. I would remove panel D, which is not informative at this initial point of the manuscript. From this “complete” overview of Juneau Icefield I would delimitate with a rectangle where panels C and E are located.

This figure is useful as it highlights the different data sources used in the publication. However, we have edited it as requested, enlarging Panel A to show in more detail the icefield, and including the coastline and ocean more clearly.

Panel A: Include boundaries for a better sea-land limit identification. **Amended as requested.**

Panel E. Some contour lines (with transparency) might help to interpret the map. **Amended as requested.**

Glacier limits in 1931 and 1908-1910 are informative, but if I am right are not used in the manuscript, and thus are not needed.

They are used to test and evidence continued glacier recession from LIA-1948. This is discussed in detail in the supplementary results.

Disconnection separation marks are barely identified in panel E. I encourage to include here not only circles with distinct colors and use triangles, squares etc...for easier identification.

The symbology here is used to be consistent with later figures. The symbology is clearly discernible on my machine and printed version. Darker colours have been used to aid clarity.

Figure 2: The number of polygons (in brackets), is not informative of the number of glaciers, I would remove this information and change it by the total extent of the ice-dammed lakes, alluvial fan,...

The number of glaciers is important; it highlights graphically the number of glaciers disappearing between the two observational periods. We have added the area of key landforms (glaciers, lakes, fans) as requested.

ELA (LIA) circles hamper the identification of the estimated LIA glacier extent. For me is more informative the LIA glacier extent (dark right) than the ELA estimation. I would remove all circles in A and include a “C pannel”(or include a new figure in the supplementary material), just including ELA (LIA) with features of different colors (with same color legend as these of the small circles). **We have removed the ELAs from the figure in the main text and added them to a figure in the SI, and referred back to this.**

Figure 3: Again, the number of polygons in brackets is not informative. Include area extent instead the number of polygons. **For clarity, the area is given in the figure, in the bottom right. There isn't enough space in the top left of the figure.**

Regarding the volume rate of change, the error must be also included. **Note that this is area change, not volume change. Following best practice we do not provide an uncertainty here.**

Disconnection and separation circles are barely seen in the figures. I encourage manuscript authors to change symbols and colours for a proper identification. **These are an important part of the story; in any case the disconnections are provided as shapefiles if needed for further clarity.**

Figure 4: Panel letters are not included (a), b), c)...) what makes difficult figure interpretation. **Panel letters were included in the y axis for each graph. However we have added them as titles for each panel to aid clarity.**

Figure 4 caption: Summer air temperatures are computed for which time period?, please, include it. **The time period is shown on the x axis of the graph.**

For panel b) is stated “The timing of an early, mid and late LIA is shown.” Where is this shown? I think this is an interpretation of the graph, at least nothing is marked in the graphs. If this sentence is an interpretation I would include it in the text not in the figure caption. **Apologies, this was an error – this refers to panel d. Amended.**

For the bottom left panel, please include a legend for the red line and the green triangle (mean and median I guess). **This is already in the caption for graph e. Clarified in the caption.**

Juneau airport data are used in the manuscript but it is not located in any figure or the distance to the study area is included. Please, provide, at least the distance from the airport to the ice field and the elevation of the weather station in the text/caption of this figure. **Juneau Airport is added to the updated Figure 5.**

Extended data 1: Please include the dates for which the ELA was computed. **These data have been added to the figure.** Hypsometric indexes are used in the manuscript? I guess these values are not used. Please, if so, remove them in this figure. **The hypsometric index is referred to in the discussion (bottom heavy versus top heavy icefields).**

Supplementary information:

Line 57 and 60. Please include precipitation in mm (International system units). **Amended.**

Line 76 There is a typo in the year. I guess it is not 2948. **Amended**

Line 156 : Where is Taku Point?? Show it or include a reference here. **We have added a new figure with these placenames all clearly marked.**

Supplementary figure 1: 0-100m and 100-200 m, ice thicknesses representation can be improved with more classes (0-50, 50-100....). This would show the areas with extreme glacier retreat in the forthcoming years. **Amended as requested. Symbology changed to show this more clearly.**

Supplementary Figure 3: Include the location of the study area in NW North America (at least a point with the location or an small rectangle). **Amended as requested.**

Supplementary Figure 5: I might remove the 5 upper glaciers in this figure as it is difficult to identify the distinct lines. From my understanding is better to show the methodology in 2-3 glaciers with good examples than showing it with 10 examples barely observed. **This figure clearly shows this in the larger Willison, Echo and other glaciers; all glaciers are included to show the diversity of glacier sizes and locations used.**

Reviewer #2 (Remarks to the Author):

This paper reconstructs ice volume and areal changes for the Juneau Icefield (JIF), a plateau icefield with outlet glaciers, and adjacent glaciers from the 19th century to 2019, using multiple data sets, including geomorphological and topographic maps, air photos and satellite imagery. The most noteworthy finding is that the rate of ice loss from the JIF has not been linear and has accelerated sharply since 2005 and that the acceleration is due, not to an acceleration in temperature increases, but is due to the JIF hypsometry, as the ELA for the icefield has now risen to the plateau surface. This means that ablation zone is now rapidly expanding over a wide area and the rate of ice loss will continue to increase even if temperature increases stay low.

These findings are of significance to glaciologists and climate and sea-level modellers for a number of reasons. Most importantly, the study suggests that current model projections of ice loss and resulting sea level rise from Alaskan and other ice fields may be underestimates, as they are based on a linear rate of ice loss. This has important implications for the rate of sea-level change in the next 100 years, as Alaskan glaciers are an important contributor in the short term.

We are delighted by these positive comments and that the reviewer recognises the significance of the work.

Secondly, the paper provides a model for the mechanisms of ice field retreat which is applicable to both modern and past ice fields globally. The paper builds on previous studies of Juneau ice field and is a significant new contribution to the literature.

Thank you for these generous comments.

The use of sequential DEMs to assess ice loss is not new, (see, for example Berthier et al., 2010 and 2018), but this approach has been previously applied to changes in this century only. The generation of a sequential DEMs covering the period from 1770-2020 is an important achievement, and is a result of combining multiple datasets, including geomorphological and topographical maps, air photos and satellite imagery, to produce the time series.

We are delighted that the reviewer recognises achievement here. Thank you for these generous comments.

The methodology used here is sound and the resulting data is of high quality; possible errors are discussed in detail in the methods and supplementary data expands on some of these data sets.

We are delighted that the reviewer recognises our method as sound and the data as high quality.

There is sufficient detail provided to reproduce the work.

Thank you for this comment.

However, I do have some issues with the conclusions drawn from the work presented. This is due more to the writing style and presentation of data within the main text than to any inaccuracies in approach. Nevertheless, some rearrangement/minor rewriting is necessary to highlight particular aspects of the data and make it clear when the authors are referring to their own or other datasets. In particular, I think the following needs addressing:

1. A key finding of the paper is that the rising ELA of the Juneau Icefield now intersects the elevations of the icefield plateau; this is stated without preamble on line 171. However, the only clear ELA data presented in the results section is for the LIA. Instead, evidence is presented for thinning across the icefield between 2010-2020, plus a large increase in disconnections in this period. However, this data is buried within a paragraph (lines 114-124) and it took me some time to figure out that this must be the supporting evidence for the statement on line 171, and that this is clearly different to previous periods. The authors should clarify if this is the case. If it is not, then they need to make clear what the evidence is for the ELA rising to the plateau height.

The evidence for the ELA rise is from the detailed literature and long mass balance records from the Icefield (McNeil et al., 2020; McNeil et al., 2019; O'Neel et al., 2019). Cumulative mass balance data is shown in Supplementary figure 2a and rising ELAs can be seen in Figure 6i. These details were reviewed in the Supplementary information, but they are an important part of the story and key evidence. We have therefore moved these details, with data on the rising ELAs and comparison with the plateau height, to the introduction of the manuscript. We have also added the mapped snowline in 2019, with comparison to the plateau, in figure 5. We have added further detail on the newly mapped snowlines to the Supplementary Information. This should clarify the ELA rise and support the interpretations more clearly.

2. Line 178 – It is not clear here whether the dramatic reduction in icefield accumulation area refers to your data or the predicted changes. I assumed on first reading it was the predicted change, as the authors have not calculated a change in accumulation area volume, so cannot say it is dramatic as yet? If this does refer to observations presented in the paper, the authors need to make it clear why specifically they think the accumulation area has decreased dramatically, as this would require retreat of the ELA beyond the edge and toward the interior of the plateau.

This is from the mass balance modelling in the paragraph above. Citation added for clarity.

3. Lines 178- 196, I think, refer to the model of ice recession presented in Fig. 6, but the absence of an initial linking statement to this effect means that it reads as if the work presented here directly measure the changes implied, e.g. snow cover duration. This fails to distinguish between the authors' own work and that of others adequately and needs addressing.

We have added a link to the figure to aid clarity.

4. I was surprised that there is no reference to the two papers by Berthier et al. (2010 and 2018) on estimating recent rates of change for Alaskan including JIF (2010) and JIF specifically (2018). The 2018 paper is mentioned in passing, but both papers show an earlier attempt to estimate ice loss using DEMs, although at a lower resolution, with much less coverage and for a much shorter time. In particular, it is worth considering a comparison of the authors' work to the 2018 Dh/Dt work on specific glaciers, to see if there is a significant change. This need not be included in the main paper, but should be in the extended methods?

These are excellent papers, and the lack of referencing was only in an attempt of brevity. Direct comparison with Berthier et al 2010 is challenging because this work only covered a small fraction of Juneau Icefield, and because the data are not publicly available for interrogation.

The comparison would rather be between the Hugonnet data (2000-2019) and the Berthier results (2000-2016), which is more of a comparison between the gaussian process registration used in Hugonnet et al and the linear regression used in Berthier et al. A different comparison would be with the 2010 paper that used DEM differencing, but challenging without having access to the original data. It is also not likely to be particularly illuminating due to difference in the methods and timeframes used.

We have added a reference to both papers in the 'Glacier mass balance' section at the paper's introduction.

There are other minor issues/typos within the figure and text to address:

We have made all of these amendments as suggested, which has improved the clarity of the manuscript. Thank you for these suggestions.

Line 58: “way a..” – remove a **Amended**

Figure 1: I didn’t find this particularly helpful, other than the map showing the location of the JIF. Consider merging 1A with Figure 3 (but see below re Figs 2-3) and moving the rest to the supplementary/extended sections. **Figure 1 has been updated.**

Line 83: I would extend this to say why Lemon Creek is used specifically – presumably because of the excellent data set? But not everyone knows that as a matter of course. **Because ELAs have been measured here (see the text moved to the introduction).**

Line 88 – worth saying “underwent dramatic changes *in area* from the LIA..” – I had to reread the paragraph to figure that out. I recommend checking throughout that you have been precise in specifying area/volume etc. – it’s quite confusing otherwise. **Clarified in text.**

Figure 2A: the ELA circles almost entirely obscure the smaller glaciers so it is hard to see the LIA extent vs the 2019 extent. You could make the circles smaller, but a better alternative might be to change the order of the figures – put Figure 3 before Figure 2 as it shows the areal extent clearly. There is also room to add a v small map showing the location of the JIF on Fig. 3. **This figure (now figure 4) has been updated and amended, so it is clearer. The reconstructed LIA ELAs have been moved to the supplementary information. The location is shown clearly in figures 1 and 2.**

Figure 2b: The figure caption vs the legend is confusing. The legend implies all the outlet glaciers except for Taku have increased in height, but I think the inverse applies? i.e. the figure caption is correct and this shows the amount of lowering, rather than the amount of change. **figure amended as requested.**

Line 100 – also refer to your Figure 3. **Amended as requested**

Line 107: grammar **amended**

Line 115: grammar – should be “...areal loss accelerated sharply...” **Amended as requested**

Line 121: you need to distinguish more clearly between the “icefield plateau” (above) and the icefield (this line) – suggest “the whole icefield”. I also got confused in places between glaciers (presumably including outlets) and glaciers (not including outlets possibly?) – maybe worth saying “all glaciers” when you mean the lot. **Amended as requested**

Lines 119-122: key points buried in the middle of a paragraph – need to be highlighted....**Have inserted paragraph break so these key points are emphasised in a new paragraph.**

Line 125: this reads as a contradictory statement – i.e. “rates of areal recession rise steadily from 1948-2005” compared to lines 107 and 128, where you mention the increase in rate from 1979-2010. Needs clarifying. **Clarified in text.**

Line 130: - do you mean ice loss? Or have glaciers completely disappeared? – if the latter, more info needed! **108 glaciers have disappeared; but this is referring to ice mass loss rather than loss of individual glaciers. Clarified in the text.**

Figure 3: Love this, but the disconnection and separation dots are a little hard to see – brighter colours needed? **Amended as suggested.**

Figure 4: It was a pain relating the captions to the graph. I suggest you rearrange the caption and lettering, going across instead of down, i.e. **We go in this order so that the time lines are the same on the up-down axes. Left column is 1650-2020 AD, right column is 1940-2020 AD. This makes the datasets easier to**

compare. We have added titles to the panels (a, b, c, d etc), which makes it easier to see and identify the panels.

a b

c d

e f etc.

Line 155: I strongly recommend making your overall point at the end of the first sentence, i.e. temperatures important, but the relationship between temperature and volume/area loss is non-linear. Otherwise this is a very disjointed read. You can still make it at the end and suggest a threshold response. We have re-ordered the paragraph as suggested.

Line 171: see my point above – you haven't drawn the links back to the data. Amended

Line 176: add "... observational data *presented here*..." – all very abstracted otherwise. Amended as suggested

Line 178: see main comments above Amended as suggested

Figure 6: This is ok, but the photograph caption is not accurate – that's not the JIF, just a small area of it. Noted. This is Vaughan Lewis Icefall. Noted in the caption.

Line 204: refer to supplementary evidence and reference 17. Reference to Davies et al 2022 included.

Line 234: grammar... Amended

Line 274: I think it would be better if 17 was not in superscript – it forms part of the main sentence, it's not the equivalent of an asterix. The same replies to references 46 and 17 below, and references in the extended data where the reference number is part of the main sentence. The citation is included now in the sentence.

Line 466: you mention that 1979 was in 4 blocks below – include here with 1948? 1979 was in one block and 1948 in 4 blocks, which is outlined in the manuscript.

Lines 587: how many are there? This information has been added in the Supplementary Information (supplementary table 8) and the text.

Line 590: This could be compared to the Berthier et al. (2018) example to see if your DEMs make a difference in calculating area/volume? As noted above, any difference would be down to the difference in regression approach, which isn't the point of the study undertaken here.

Reviewer #3 (Remarks to the Author):

The authors present a compilation of observations and reconstructions of changes of the Juneau Icefield that is probably unique for a glacierized mountain chain of this scale in its spatial and temporal scope, and in its comprehensiveness.

Thank you for this positive and supportive comment.

Similar densities of observations are found, if at all, in regions like the Alps or Scandinavia, which arguably are less relevant than Alaska to many glaciological research questions. These other data sets have not been created because of their urgency, but because of convenience and traditions. The authors therefore deserve a lot of praise for their effort to improve the knowledge in a region that is much more relevant and much harder to investigate, and they have obviously succeeded in connecting the different qualities, quantities

and characteristics of their data to build a panoramic view of the evolution of the Juneau Icefield since the Little Ice Age.

Thank you for these very positive and generous comments and we are delighted that the reviewer acknowledges the value of the dataset.

There are therefore very strong arguments that this work should be published; it would be of high value to the glaciological research community (and probably beyond).

Thank you for this positive and generous comment.

However, I see two fundamental problems in the manuscript that are not related to the data, but to their interpretation and the framing of the entire manuscript.

(1) The smaller of these problems concerns the discussion of the relevance of the findings with respect to glacier models and projections of glacier evolution. The manuscript claims that “Current analyses of future glacier behaviour predict a linear future contribution of global (8) and Alaskan ice masses to sea level rise (5,6,8), and a linear rate of volume loss for Juneau Icefield (9).”, and that “the simplified physics deployed by many of the models in Glacier MIP [...] accounts poorly for variation across different hypsometries (11,14)”.

The manuscript argues that the nonlinearities supposedly found in the data (see below for more on this) cannot be captured by the current generation of models. I strongly disagree with this assessment. The given references (5, 6, 8, 9) show nonlinearities in the mass change rates over the 21st century, such that invoking “such linear projections” does not adequately reflect their characteristics. It is true that regional and global scale projections “hide” nonlinear behavior of individual glaciers in the aggregated numbers typically shown. This does not mean, however, that this kind of behavior is not represented at the scale of individual glaciers (and it even shows in the aggregate, as stated above). In particular, the well-known feedbacks between glacier hypsometry/geometry and mass balance referred to in the manuscript are well represented in state-of-the-art global glacier models as used in references 6, 8, and 9. Some other nonlinearities are possible in some, but not all of the global-scale models (e.g., concerning disconnections). There are additional nonlinearities which are not captured well by any of this type of model (the manuscript mentions some), these are addressed, e.g., in ref. 14. But those nonlinearities are not the ones the manuscript claims to be affecting the Juneau Icefield.

We have refocused the manuscript’s motivation and conclusions to focus more on the empirical observations and have removed focus on the limitations of current numerical modelling attempts. While existing numerical simulations forecast a linear rise in rates of volume loss (cf. Rounce et al. abstract, “mass loss is linearly related to temperature increase”), even using PISM (Ziemen et al., 2016), and OGGM, which can simulate disconnections and more of the full processes, and while many of the other models in GlacierMIP are vastly simpler, some relying on just processes like volume-area scaling (Marzeion et al., 2020), we argue that the story presented by the impressive acceleration of Juneau Icefield is alone sufficiently interesting. We therefore simplify our introduction and discussion by focusing less on model limitations and more on our own empirical data.

This issue does not affect the center of the manuscript, and because of this, I see this as the “smaller” of the two fundamental problems. I see it as fundamental since it affects a main motivation of the manuscript, being prominently given in the abstract (and repeatedly mentioned in the manuscript). It should be possible to reframe the manuscript so that it is more adequately referring to the state-of-the-art in glacier modeling.

We hope that since we have removed this from the top and tail of the manuscript and leave the motivation as requiring longer term studies to understand in more detail the trends in area and volume change. we

have therefore reframed the manuscript.

(2) The bigger of the two problems concerns the claim that nonlinear processes are driving the mass loss acceleration of the Juneau Icefield (see title). The manuscript has not convinced me that there is in fact a nonlinear response of the icefield observable in the data. Essentially, this conclusion would have to rest on the scatter plot in Fig. 4j. Here, a nonlinear response of the rates of area and thickness change to temperature change would have to be visible. With only 4 (thickness) and 6 (area) point available, and quite a bit of scatter (which might be explained by precipitation anomalies), the argument for nonlinearity is weak. The statement in lines 165 – 168 is therefore not justified; neither the part concerning the present manuscript, nor concerning the modeling studies (see above). Arguing the other way around (“nonlinear responses found in projections are not yet robustly detected in observations/reconstructions”) would be a similar misstatement, but arguably not worse than what is said in the manuscript.

Just to note that the scatter is caused by precipitation anomalies.

As an alternative to showing a nonlinear response of the entire icefield (Fig. 4j), it would be possible to demonstrate that nonlinear effects are affecting the glaciers by showing that the ELA has crossed into the plateau region (i.e., by looking at ELA changes and how they relate to the hypsometry of the glacier), or by investigating the mass balance profile and ice flow velocities in the region of glacier disconnections (see explanation of feedbacks around Fig. 6). Unfortunately, this is not done in the manuscript.

The non-linear effects are indeed driven by the ELA crossing the plateau, which we had not adequately demonstrated in the previous version of the manuscript. The review of the ELA was rather buried in the Supplementary Information. We have moved the description of the vast previous work on Juneau Icefield ELAs (McNeil et al., 2020; McNeil et al., 2019; McNeil et al., 2016) and snowlines (Pelto, 2019) to the main text. Further to this, we have built on previous late-summer snowline mapping (Pelto, 2019) by analysing the 2019 late-summer snowlines; see new table in Supplementary Information and addition to figure 5. These ELA changes show the rising ELAs and their relationship to glacier hypsometry, with observed summer snowlines high on the plateau in 2019.

In terms of mass balance profiles at the site of disconnections, our data do show increased thinning at the mean height of the disconnections. Firstly, figure 6i shows that the rising ELAs have reached the mean elevation of the disconnections, resulting in increased numbers of observations, as seen in figure 6h. secondly, the elevation at which thinning occurs rises through the observational period (as shown in figure 7). We have strengthened the discussion of these results under ‘Icefield fragmentation’ in the Results.

We hope that this has strengthened our identification of the non-linear processes affecting the icefield that are outlined in Figure 8.

In this context, the section “Hypsometric controls of icefield behavior” is very problematic: the relevant mechanisms are well-known, but they are also explained (and referenced) well in the manuscript. However, the manuscript does not show that a single one of them is actually affecting the Juneau Icefield in their data. Remarkably, they use an edited photograph (Fig. 6) for illustrating “how it might look in a few decades”. That is exactly the point: in this manuscript, I don’t find any evidence that these feedbacks are active in present day. However, the manuscript very boldly claims that it is the case in the title, the abstract, scattered in the text, and in the final, conclusive paragraph.

We argue here that the hypsometry of the icefield is driving the acceleration of thinning across the plateau. As suggested by the reviewer above, here we again strengthen the argument by referring back to the observations of ELA and snowlines on the plateau, resulting in the processes and feedbacks identified in figure 8 (previously figure 6). This includes the reduced accumulation area and changed albedo as a result of

the loss of high-albedo snow on the plateau – this is mapped in figure 5. Thinning and SMB feedback is then an inevitable response to the thinning already observed on the plateau. We have developed this section of the discussion to emphasise the evidence available for each of the processes named in the blue box in figure 8 (the schematic).

We also show increasing icefield fragmentation as glaciers disconnect, which increases bare rock as the icefield fragments, driving albedo changes, increasing debris cover, and changing ice flow. We also show increased thinning on glacier tongues. Our data therefore does show that these processes are affecting Juneau Icefield.

The problem of a missing demonstration of nonlinearity in the data may arise from a confusion of “nonlinearity” and “acceleration”. E.g., an acceleration in mass change can be a linear response to an increased temperature anomaly. The feedbacks explained in the second and third paragraph of the manuscript (and Fig. 6) would lead to an acceleration in mass change without additional changes in the temperature anomaly.

And this is indeed what we have shown here.

Therefore, I would suggest that the authors carefully decide whether they actually want to claim (and then have to demonstrate) a nonlinear response of the Juneau Icefield to climate change (which, with the data on hand, seems to be very difficult), or whether they wish to make a case for an impressive acceleration of glacier change in Alaska in the recent decades, which would be a very strong case.

Thank you for this constructive comment. We agree that our work shows an acceleration in mass change due to a steadily increasing temperature forcing. The hypsometrically controlled melt-accelerating feedbacks identified in the discussion have driven this impressive acceleration of melt. These feedbacks contribute to a hysteresis in the system, that will impede glacier regrowth in event of cooling. We have updated and reframed the abstract, discussion and Figure 8 (the schematic) to better reflect these findings, and focused on acceleration rather than ‘nonlinearity’. We therefore focus on making a very strong case for the impressive acceleration of glacier change in Alaska in recent decades.

Either way, very substantial work on the text (and potentially, the data analysis) would be needed, so that I would probably consider the manuscript as “new”. This would imply a suggestion to reject the manuscript. However, as I am convinced that the work definitely should be published eventually, I would also agree with considering this as a case of “major revisions”. Expecting the manuscript to go through another round of revisions (potentially in a different format, pending the editor’s decision), I have refrained from specific/minor comments on all parts of the manuscript except the main text, where I hope also these specific comments will be helpful (also, I don’t see any major issues in the methods).

Thank you for these constructive comments, which we feel have improved the manuscript. We have reframed the discussion and abstract to focus on the observed accelerated mass change driven by hypsometrically controlled feedbacks, and have provided substantial new data and analysis on the ELAs and snowlines which are now intersecting the plateau.

Specific/minor suggestions for the main text:

L18: „The majority...” would imply > 50 %, which is not the case according to Hugonnet et al. (<https://doi.org/10.1038/s41586-021-03436-z>); perhaps change to: “The largest regional contribution...”
Amended.

L19: “...projected to become even more dominant...” I’m not sure I would agree, and would expect that this depends on the scenario: in a high-emission scenario, e.g., I would expect that regions that have more ice left (peripheries of Greenland and Antarctica, the Canadian Arctic) become more dominant over time.

Alaska is the largest regional contributor to global mean sea level from 2015-2100 (Rounce et al., 2023). The Greenland periphery, Antarctic etc, contribute all less. Citation added to the introduction (abstracts are unreferenced).

L22: “...with...” since this refers to the acceleration (but no acceleration value is given), perhaps change to: “...leading to...” Amended as suggested.

L35: suggest to change to: “Under the higher-emission scenario RCP 4.5...” Amended as suggested.

Fig. 1 is a bit disorganized: e.g., I can’t find any ice-dammed lakes, lakes, or sandurs on the map (or photographs), features not visible have been removed from the legend.

and referring to north- and south-flowing glaciers in panel d, without including a north arrow, is not ideal (perhaps label the glaciers in panel d directly?). North arrow added to figure. All maps are on the same projection.

I would also expect that the individual glaciers mentioned in the text (e.g., lines 83-87) appear in the map, or vice versa. Please include a reference to (e.g.) Fig. 2a in the relevant sections of the text. We have added key glacier names to Figure 1 to aid clarity in the manuscript.

L82-90: please indicate the uncertainties of the assessed changes in ELA, volume, and thickness. These uncertainties are important for understanding the significance of the changes. Amended as requested.

Fig. 2: I appreciate the difficulty to put the rich data set on the map(s), and unfortunately, I don’t have a good solution. But it is problematic that (i) the circles of the ELA estimates cover a large fraction of the trimline and moraine markers and (ii) moraine polygons are shown gray on gray background. In line with other reviewers, we have redesigned this figure and moved the ELA data to the SI. Constraints on numbers of display items make showing all the data challenging.

L118: what do “M” and “SD” mean? What units are used here? Mean and standard deviation; clarified in the text.

L129: It has not been shown that the acceleration is really exponential; I would suggest to state this more carefully (also in the abstract and other places in the manuscript). Amended to ‘steep’ throughout

Fig. 3: The tables in the lower left should either really be formatted as tables (outside of the figure) or left out (they are already shown in Fig. 4). Removed from figure as suggested. They’re also shown in the table moved from the SI to the main text.

Regarding number of glaciers: how are separations treated? Do they increase the number of glaciers? In line with inventory best practice, they are retained as a single glacier but are a multipart polygon. If just the terminus is separated but they still share an accumulation area, then they remain as one single polygon.

Fig. 4a: please explain red line here, not just in panel f. Amended as requested. b: Where is the timing of an early, mid and late LIA shown? This was an error - this should refer to panel d. Amended. Last sentence of the caption: what does this refer to? Panel f does not show reduced precipitation during 2005 – 2015? Removed from caption.

Fig. 5: legend and caption: please correct from dhdt to dh/dt. **Amended as requested.** More generally, I would suggest to move this figure to the supplement; it is not that central to the arguments in the manuscript. **We argue that this figure is crucial as it demonstrates some of the key arguments in the text; this includes the thinning observed across the plateau, and also the relationship between the rising elevation of the thinning and the increasing frequency of observations of glacier disconnections.**

REFERENCES

- Carrivick, J.L., Boston, C.M., Sutherland, J.L., Pearce, D., Armstrong, H., Bjørk, A., Kjeldsen, K.K., Abermann, J., Oien, R.P., Grimes, M., 2023. Mass loss of glaciers and ice caps across Greenland since the Little Ice Age. *Geophysical Research Letters* 50, e2023GL103950.
- Davies, B., Bendle, J., Carrivick, J., McNabb, R., McNeil, C., Pelto, M., Campbell, S., Holt, T.O., Ely, J.C., Markle, B.R., 2022. Topographic controls on ice flow and recession for Juneau Icefield (Alaska/British Columbia). *Earth Surface Processes and Landforms* 47, 2357-2390.
- Marzeion, B., Hock, R., Anderson, B., Bliss, A., Champollion, N., Fujita, K., Huss, M., Immerzeel, W., Kraaijenbrink, P., Malles, J.-H., Maussion, F., Radić, V., Rounce, D.R., Sakai, A., Shannon, S., van de Wal, R., Zekollari, H., 2020. Partitioning the Uncertainty of Ensemble Projections of Global Glacier Mass Change. *Earth's Future* 8.
- McNeil, C., O'Neel, S., Loso, M., Pelto, M., Sass, L., Baker, E.H., Campbell, S., 2020. Explaining mass balance and retreat dichotomies at Taku and Lemon Creek Glaciers, Alaska. *Journal of Glaciology* 66, 530-542.
- McNeil, C.J., Campbell, S.W., O'Neel, S., Baker, E.H., 2019. Glacier-Wide Mass Balance and Compiled Data Inputs: Juneau Icefield Glaciers (ver. 4.0, December 2022). U.S. Geological Survey data release. US Geological Survey, Alaska Science Center.
- McNeil, C.J., Sass, L.C., Florentine, C.E., Baker, E.H., Peitzsch, E.H., Whorton, E.N., Miller, Z.S., Fagre, D.B., Clark, A.M., O'Neel, S., 2016. Glacier-Wide Mass Balance and Compiled Data Inputs: USGS Benchmark Glaciers (ver. 7.0, December 2022): U.S. Geological Survey data release.
- O'Neel, S., McNeil, C., Sass, L.C., Florentine, C., Baker, E.H., Peitzsch, E., McGrath, D., Fountain, A.G., Fagre, D., 2019. Reanalysis of the US Geological Survey Benchmark Glaciers: long-term insight into climate forcing of glacier mass balance. *Journal of Glaciology* 65, 850-866.
- Paul, F., Bolch, T., Briggs, K., Kääb, A., McMillan, M., McNabb, R., Nagler, T., Nuth, C., Rastner, P., Strozzi, T., Wuite, J., 2017. Error sources and guidelines for quality assessment of glacier area, elevation change, and velocity products derived from satellite data in the Glaciers_cci project. *Remote Sensing of Environment* 203, 256-275.
- Pelto, M., 2019. Exceptionally High 2018 Equilibrium Line Altitude on Taku Glacier, Alaska. *Remote Sensing* 11, 2378-2378.
- Rea, B.R., 2009. Defining modern day Area-Altitude Balance Ratios (AABRs) and their use in glacier-climate reconstructions. *Quaternary Science Reviews* 28, 237-248.

Rounce, D.R., Hock, R., Maussion, F., Hugonnet, R., Kochtitzky, W., Huss, M., Berthier, E., Brinkerhoff, D., Compagno, L., Copland, L., Farinotti, D., Menounos, B., McNabb, R.W., 2023. Global glacier change in the 21st century: Every increase in temperature matters. *Science* 379, 78-83.

Ziemen, F.A., Hock, R., Aschwanden, A., Khroulev, C., Kienholz, C., Melkonian, A., Zhang, J., 2016. Modeling the evolution of the Juneau Icefield between 1971 and 2100 using the Parallel Ice Sheet Model (PISM). *Journal of Glaciology* 62, 199-214.

Reviewer #1 (Remarks to the Author):

Review of the manuscript "Accelerating glacier volume loss on Juneau Icefield driven by hypsometry and melt-accelerating feedbacks" by Bethan Davies et al.

I appreciate the changes included in the manuscript by the authors and I think their work has now been improved. Nonetheless from my point of view there is still work to do. And I recommend major review accomplishing these important points (the first of them not demonstrated in the manuscript).

1. Despite I agree with the hysteresis behavior of glaciers and that Juneau icefield regrowth is challenged due to the hypsometrical changes of the icefield, I consider that all the feedbacks have only been named (yes, with suitable references) but not shown with observations and data in the manuscript. This way, I consider it is not worth to claim about which are the main drivers of the acceleration. The authors clearly show the acceleration of the area and volume loss but do not analyze what is causing this acceleration (the only data shown in this regard is temperature rise). Such strong assertion (what causes the acceleration), needs to be verified with data (albedo trends, mean slope evolution, debris cover extent trend, ice flow evolution...). Otherwise, the title and the main conclusions must be consequently changed. Somehow, this is related with Reviewer 3 remarks. I completely agree about what he/she points out: "I don't find any evidence that these feedbacks are active in present day". From my perspective, the response of the authors and the changes included in the text do not answer this lack. No albedo change is shown, no debris cover increase is shown, no iceflow change is shown through the study period. I consider the manuscript must be conveniently changed in this regard before publications. For example, lines 433-435 should be removed and also some sections in the discussion. The named feedbacks can be described or presented as an hypothesis, but not as a trend observed. In this sense, I agree on the way of presenting them in lines 376 and 377: "...feedbacks CAN lead to an increase". Overall, the melt accelerating feedbacks can be discussed, but one schematic figure (figure 8) do not justify these are driving (and will drive) Juneau Icefield volume loss acceleration and that now a "tipping point is reached controlled by a number of feedbacks" (stated in the abstract) as these have not been shown in the article.

2. One potential way to show if a tipping point has been reached could be to show the ELA evolution in the last years. In lines 269-276, the manuscript shows the ELA observed in 2019. Why the ELA mean elevation is not updated until 2023? Sentinel images are available until yesterday (literally) and this will definitely show if the ELA is rising in the last years and area above/below the plateau mean elevation is being reduced. Please, update this information to firmly show (or not) that this accelerating trend is being observed nowadays. I consider this is a relevant point, update the entire database of volume and area losses would be time consuming but Sentinel derived ELAs (potentially 2022 and 2023) is much faster to derive and will help to see the recent evolution of Juneau icefield.

I still have a list of minor comments to be addressed:

Line 247. Volume loss of $0.65 \pm 0.92 \text{ km}^2 \text{ a}^{-1}$ means that, a gain of $0.27 \text{ km}^2 \text{ a}^{-1}$ can be a true value during the study period. Such a big error compared to the mean value, is related to the method used to derive it or is it related to the interannual variability of glaciers mass balance? I think it is worth to comment this point.

Figure 1. Please include a coast limits with a black line. Change the color to represent the ocean/water, as in low elevation areas it is not possible to differentiate land and ocean. A small location map (of north America which is easily recognizable) is needed in Figure 1 (it can be included in the upper right area of the upper panel). I think it is not needed to include all area vs elevations histograms for Alaskan ice fields. The cumulative area vs cumulative elevation is informative enough. I would only include the Juneau area elevation histograms.

Figure 2: Lower right map. Symbols are difficult to see. Make them bigger and change colors (specifically disconnection in panel E have same color as the LIA extent and this is confusing).

Figure 3: Almost impossible to identify (in printed version and online version) disconnection and separations with the symbols and colors used. Maybe there are not disconnections or separations in

these areas, if this is the case, do not include these symbols in the legend please. Change conveniently.

Figure 6 a): The temperature anomaly is computed for which reference time period? Please include it in the text

Figure 4: answer to reviewer 1 comment on figure 2: Yes number of glaciers is important but I guess features are not in all cases glaciers. Clearly indicate that the number of features (in brackets) is not glacier number. I appreciate that the area has been included in the new version, this is informative.

Figure 4: In the LIA area and glacier extent is not needed to include decimals if the error is three others of magnitude larger than the decimals included. Remove them please. Same for decimals in lines 223, 226, 259, Table 1 area changes, table 2 total volume loss, etc.... The 50 km scale bar is confusing. Edit it including smaller divisions (as those included in figure 5 or 7).

Figure 4 caption. For clarity, include the period for which the ice surface lowering has been computed.

Figure 5: Same comment of Figure 3. Disconnection symbols are difficult to identify. I would change colors. Also, I encourage to make more evident the difference between 1200 and 1500 m plateau elevations. In both printed version and in screen version, is barely observed the transition.

Figure 7: Include the number of disconnections registered in each time period please.

Supplementary figure 4: Despite in the reviewers' response it is said that the location of the study area is included, it is not. Please mark in all ERA5 reanalysis maps where Juneau icefield is located. Also include a legend for it.

Response to points 2 and 3 (Lines 178 and 178-196 respectively) of reviewer 2: Please include more details on how these points have been addressed, also including the changes done in the text and why you consider is appropriate.

Reviewer #2 (Remarks to the Author):

This is my second time reviewing this article and I am happy that all my suggestions for improvement have been incorporated. As I stated previously, this is a significant piece of work, based on extensive, thoroughly analysed datasets, which support the conclusion and claims. The rearrangement and rewording of the paper has highlighted the significant key findings.

A couple of typos:

Line 110: should be "in the same time period" or "at the same time"

Line 137/8: a 95% confidence interval, rather than the 95%...?

Line 226: "... the icefield is 315." rather than "of"

Reviewer #3 (Remarks to the Author):

I would like to thank the authors for having done a great job revising the manuscript following (or, where appropriate, arguing against) the reviewers' comments, and I am very happy with the overall quality of the manuscript! Now even more than before I am convinced that these results should be published, and I see less issues with how some of the results are presented and discussed. The manuscript also reads very well and the figures have been improved substantially, so that I have hardly any minor suggestions. But I still see two points that I would strongly suggest to be changed before publication.

The first concerns the claim of a tipping point being reached. The manuscript makes now a much better case arguing that the tipping point is in reach (i.e., that ELAs are approaching the plateau elevations, and are crossing them in individual years), but some of the statements still are too bold. Mostly, I'm concerned with the last sentence of the abstract ("This icefield has now reached a tipping point controlled by a number of feedbacks, with hysteresis inhibiting glacier regrowth."). This is a categorical statement, and while I would agree that it seems very likely that this threshold will be

crossed (and in fact might be, and interannual variability clearly already pushes the ELA above threshold elevations), I don't think the claim that the hysteresis is already affecting the SMB is acting right now, is substantiated as clearly in the manuscript as the abstract makes believe. In fact, within the main text, the statements are a lot more careful ("...a decrease in height of a flatter icefield or ice cap will theoretically result in a decrease in surface mass balance (SMB) as the ice surface lowers into warmer air, which reinforces itself as an SMB-elevation positive feedback", or: "the mass balance-elevation feedback will inhibit glacier regrowth (cf.15), likely forming an irreversible tipping point in glacier dynamics"). The melt-elevation feedback is potentially relatively slow. If I compare the rate at which the ELA is rising (from Fig 6i I guess ~200 m over the past 20 years?) to the thinning rate of the plateau (hard to guess from Fig. 7, but from Fig. 6c it should be less than 1 m/yr in recent years), this would indicate that the acceleration in mass loss is clearly dominated by the rise of the ELA (this agrees well with the sentence starting in L356), not by the surface elevation-melt feedback (it might be informative to calculate these rates and compare in the manuscript). I come away with the impression that the authors are looking for a dramatic statement for the abstract and are overshooting – the results they present in the main body of the manuscript are dramatic enough. I simply suggest somewhat more careful wording in the abstract.

The second is still concerning the edited photograph (now Fig. 8). In the end, this is an editorial decision, but I would really strongly advise against publishing an edited photo this way, which to superficial readers (and let's be honest, there are more than enough) may be mistaken as a real one. Unfortunately, in climate science, we additionally have to consider purposeful misunderstandings, and I would detest seeing this figure pop up in social media with the claim that if glaciologists don't find evidence in the field, they resort to editing photos. The relevant points of the figure could all be made in a schematic drawing of the landscape, and I really don't see any advantage of using an edited photo over a drawn illustration for explaining the concepts of the relevant processes.

One minor point: Please check the legend of Fig 3, there is a double entry (sandur) and I'm not sure all the features in the legend actually occur in the maps (I don't see disconnections and separations).

Response to reviewers

Reviewer #1 (Remarks to the Author):

Review of the manuscript “Accelerating glacier volume loss on Juneau Icefield driven by hypsometry and melt-accelerating feedbacks” by Bethan Davies et al.

I appreciate the changes included in the manuscript by the authors and I think their work has now been improved. Nonetheless from my point of view there is still work to do. And I recommend major review accomplishing these important points (the first of them not demonstrated in the manuscript).

1. Despite I agree with the hysteresis behavior of glaciers and that Juneau icefield regrowth is challenged due to the hypsometrical changes of the icefield, I consider that all the feedbacks have only been named (yes, with suitable references) but not shown with observations and data in the manuscript. This way, I consider it is not worth to claim about which are the main drivers of the acceleration. The authors clearly show the acceleration of the area and volume loss but do not analyze what is causing this acceleration (the only data shown in this regard is temperature rise). Such strong assertion (what causes the acceleration), needs to be verified with data (albedo trends, mean slope evolution, debris cover extent trend, ice flow evolution...). Otherwise, the title and the main conclusions must be consequently changed. Somehow, this is related with Reviewer 3 remarks. I completely agree about what he/she points out: “I don’t find any evidence that these feedbacks are active in present day”. From my perspective, the response of the authors and the changes included in the text do not answer this lack. No albedo change is shown, no debris cover increase is shown, no iceflow change is shown through the study period. I consider the manuscript must be conveniently changed in this regard before publications. For example, lines 433-435 should be removed and also some sections in the discussion. The named feedbacks can be described or presented as an hypothesis, but not as a trend observed. In this sense, I agree on the way of presenting them in lines 376 and 377: “...feedbacks CAN lead to an increase ...”. Overall, the melt accelerating feedbacks can be discussed, but one schematic figure (figure 8) do not justify these are driving (and will drive) Juneau Icefield volume loss acceleration and that now a “tipping point is reached controlled by a number of feedbacks” (stated in the abstract) as these have not been shown in the article.

As also suggested by Reviewer 3, we have rephrased the abstract (please see comment to reviewer 3 below) regarding the tipping point in the abstract, in order to be more clear about what the article data shows:

“Rising equilibrium line altitudes and increasing ablation across the plateau has driven a series of hypsometrically controlled melt-accelerating feedbacks and resulted in the observed acceleration in mass loss. As glacier thinning on the plateau continues, a mass balance-elevation feedback is likely to inhibit future glacier regrowth, potentially pushing glaciers beyond a dynamic tipping point.”

We have undertaken new work to show both snowline and albedo change across multiple decades, and present this new data in the Supplementary information. We have also brought the snowline and albedo data together under a new heading in the Results section, to emphasise the new data and to differentiate it from the glacier area and volume change. We believe that this strengthens our argument in the manuscript. In the Results and from published datasets, we have therefore more clearly demonstrated that the ELA and

snowline are rising to intersect with the plateau, the albedo is decreasing, and document the fragmentation of glacier ice and the diminishing ice cover due to glacier recession.

In the discussion, we are careful to, as the reviewer suggests, present the feedbacks as a hypothesis, as we agree that we have not demonstrated albedo change (bold text denotes the changes):

“We hypothesise that the dramatic reduction in icefield accumulation area **is decreasing** overall icefield albedo. As snow cover duration and extent is reduced (cf. Figure 5), darker rock, firn and ice surfaces are **increasingly** exposed, which absorb more solar radiation, enhancing the ablation rate. This albedo feedback (cf.^{1,2} and illustrated in Figure 8) **is likely to** then contribute to further icefield thinning. This effect is compounded by deposition of black carbon³ and dust⁴, which is set to increase in future, due to a rise in tourism and wildfires⁵.

Thinning on **low-slope icefield plateaux**, as observed in Figure 7, **is likely to** drive an elevation-SMB feedback by lowering the icefield surface and driving increased melt in warmer air temperatures (due to the lapse-rate induced rise in air temperature⁶⁻⁹) (**Figure 8**). This response is fundamentally unstable, as thinning drives increasingly negative mass balances, even in the absence of further climate forcing⁹. In some places across the **Juneau** plateau, the ice is over 600 m thick¹⁰ (Supplementary figure 2), so the ice-elevation feedback has a large range over which to operate. This topographic feedback can lead to an increased frequency of extreme negative mass balance rates, especially when the climate data diverges from the mean⁶. **Once initiated**, long-term thinning is inevitable, because even in the event of climate stabilisation, the mass balance-elevation feedback will inhibit glacier regrowth (cf.¹¹), likely forming an irreversible tipping point in glacier dynamics.”

With regards to debris cover, a statistical relationship between glacier disconnection and glacier debris cover was demonstrated in a previous publication by these authors, which is cited here. We have added the citation more clearly when needed. We have also been more careful to cite classic papers demonstrating the influence of debris on glacier melt (e.g., ¹²) (changes in bold):

“Thin debris accumulates on the ice surface, darkening the surface and thereby enhancing melt. In some places, debris accumulates to the point that the glacier becomes debris-covered (>50% of the tongue is covered in thin debris¹³). **Thicker debris cover on glacier ice encourages *in situ* down-wasting** but slows rates of glacier area loss **due to insulating impact of debris cover on surface melt**¹². The proportion of debris-covered glacier area to clean-ice glacier area will likely increase as glacier shrinkage continues, increasing the relative importance of debris cover.” ... **“We hypothesise that** increased bare rock exposure and the loss of ice **within and around glaciers** will act to darken the region, reducing regional albedo, which may contribute to further regional warming.”

We therefore have, as the reviewer suggests, taken care to ensure that the feedbacks are discussed and presented as a hypothesis, and not an observation.

Under the section headed ‘Wider Implications’, we have amended lines 433-435 to remove ‘as our observations demonstrate’. These feedbacks are well understood and are cited in the article extensively so this sentence is justified.

2. One potential way to show if a tipping point has been reached could be to show the ELA evolution in the last years. In lines 269-276, the manuscript shows the ELA observed in 2019. Why the ELA mean elevation is not updated until 2023? Sentinel images are available until yesterday (literally) and this will definitely show if

the ELA is rising in the last years and area above/below the plateaus mean elevation is being reduced. Please, update this information to firmly show (or not) that this accelerating trend is being observed nowadays. I consider this is a relevant point, update the entire database of volume and area losses would be time consuming but Sentinel derived ELAs (potentially 2022 and 2023) is much faster to derive and will help to see the recent evolution of Juneau icefield.

There is already an extensive and long-term monitoring dataset over Juneau icefield, which shows the rising ELAs which are intersecting the plateau. This is shown for example in figure 6i – here, the measured ELAs for two glaciers and the height of the plateau is emphasised. You can clearly see the rising ELAs intersecting the plateau. Snowline elevations are a proxy for these more detailed datasets. There are also long-term datasets measuring transient snowlines on Taku Glacier from Pelto 2019¹⁴ which support this analysis.

In order to thoroughly address this point, we mapped new end of summer snowlines from 2020-2023, meaning that there is a new icefield wide dataset from 2019-2023. These data are shown in detail in the revised Supplementary Information section 5.6, with updated tables and maps showing the snowline elevation. We also updated the table of snowline information in the Source Data. These new datasets show that snowlines have continued to reach the plateau in recent years, though 2019 was an unusually high year.

We have supported these new observations by comparing these icefield wide snowline datasets to the long record of glacier mass balance recorded by the USGS at Lemon Creek Glacier and Taku Glacier, with new text and two new figures added to the Supplementary Information:

“Supplementary figure 14a shows the elevations of the icefield wide mapped end-of-summer snowlines in comparison with mapped snowlines from Taku Glacier and Lemon Creek Glacier from 1946-2023 (Supplementary figure 14b), using an updated dataset from Pelto (2019)¹⁴, and measured equilibrium line altitude for these two glaciers over the same time period, using data derived from the USGS Benchmark Glacier Program¹⁵; see also reference publications^{16,17} (Supplementary figure 14c). These datasets show that, although the year 2019 was an unusually high year for snowline elevation, snowlines are reaching the plateau frequently, with both snowlines and ELAs occurring above the plateau height of 1200 m since 2005.”

We have also updated the main manuscript to note this important point, with snowlines and ELAs occurring on the plateau regularly over the last 15-20 years.

We believe that this change has strengthened the argument in the manuscript and does indeed show that the icefield has reached a threshold, with increasingly high snowlines driving change.

I still have a list of minor comments to be addressed:

Line 247. Volume loss of $0.65 \pm 0.92 \text{ km}^2 \text{ a}^{-1}$ means that, a gain of $0.27 \text{ km}^2 \text{ a}^{-1}$ can be a true value during the study period. Such a big error compared to the mean value, is related to the method used to derive it or is it related to the interannual variability of glaciers mass balance? I think it is worth to comment this point. The large uncertainty here is due to the interpolation of the datasets over areas with no data. The LIA DEM is calculated only over the icefield ablation areas, and the 1948 DEM is likewise not available over Canada. This does not mean that the change is insignificant, but we emphasise that the uncertainty is large because of interpolation, not because of variability of the elevation differences.

We have added a note to the Methods to clarify this point, as written above.

Figure 1. Please include a coast limits with a black line. The complexity of the coastline makes this very difficult to show at this scale; it ends up dominating the figure. Instead, we have added a colour to the ocean, which makes the land stand out much more.

Change the color to represent the ocean/water, as in low elevation areas it is not possible to differentiate land and ocean. See above comment. We have coloured the ocean to better differentiate the water and land.

A small location map (of north America which is easily recognizable) is needed in Figure 1 (it can be included in the upper right area of the upper panel). Amended as requested.

I think it is not needed to include all area vs elevations histograms for Alaskan ice fields. The cumulative area vs cumulative elevation is informative enough. I would only include the Juneau area elevation histograms. We think that these histograms illustrate the argument well – the cumulative graphs alone are difficult to interpret.

Figure 2: Lower right map. Symbols are difficult to see. Make them bigger and change colors (specifically disconnection in panel E have same color as the LIA extent and this is confusing). The symbols were enlarged to improve their visibility. The colour is different to the LIA and this is now clearer; they are a unique colour.

Figure 3: Almost impossible to identify (in printed version and online version) disconnection and separations with the symbols and colors used. Maybe there are not disconnections or separations in these areas, if this is the case, do not include these symbols in the legend please. Change conveniently. Thanks for picking up the error – the disconnections were not layered appropriately in the GIS table of contents. This is now corrected. The symbols were also enlarged to improve their visibility.

Figure 6 a): The temperature anomaly is computed for which reference time period? Please include it in the text. The anomaly is computed against the year 1000-1998 mean, as done in the original publication (see y axis for the panel a). In panel f, the temperature anomaly is computed against the 1986-2005 mean, as is commonly used for the IPCC. This is now noted additionally in the caption.

Figure 4: answer to reviewer 1 comment on figure 2: Yes number of glaciers is important but I guess features are not in all cases glaciers. Clearly indicate that the number of features (in brackets) is not glacier number. Noted in the legend.

I appreciate that the area has been included in the new version, this is informative. Thank you.

Figure 4: In the LIA area and glacier extent is not needed to include decimals if the error is three orders of magnitude larger than the decimals included. Remove them please. Same for decimals in lines 223, 226, 259, Table 1 area changes, table 2 total volume loss, etc.... Amended to give results to 1 d.p.

Table 1 is now given to 1 d.p. for total glacier area, mean glacier area and change in area.

However, in Table 2 I think that for mean area-averaged dh/dt , two decimal places are needed due to the smaller numbers in the confidence interval.

The 50 km scale bar is confusing. Edit it including smaller divisions (as those included in figure 5 or 7). Amended as requested.

Figure 4 caption. For clarity, include the period for which the ice surface lowering has been computed. Amended as requested.

Figure 5: Same comment of Figure 3. Disconnection symbols are difficult to identify. I would change colors. The colours are unique – no other colours are really available to enhance the change. Instead I have enlarged the disconnection symbols to make them more visible.

Also, I encourage to make more evident the difference between 1200 and 1500 m plateau elevations. In both printed version and in screen version, is barely observed the transition. We have darkened the plateau colours to enhance visibility.

Figure 7: Include the number of disconnections registered in each time period please. Amended as requested. However, note that as the date of disconnection was calculated using the satellite imagery, the timings do not exactly match.

Supplementary figure 4: Despite in the reviewers' response it is said that the location of the study area is included, it is not. Please mark in all ERA5 reanalysis maps where Juneau icefield is located. Also include a legend for it. The location of Juneau is already shown in this figure, with a yellow star in panel A. This is noted in the caption. To emphasise this further we have moved the note on this to the second sentence of the caption and added a further label to panel A.

Response to points 2 and 3 (Lines 178 and 178-196 respectively) of reviewer 2: Please include more details on how these points have been addressed, also including the changes done in the text and why you consider is appropriate.

We welcome the opportunity to provide more information here.

Lines have now changed as extensive information was moved from the Supplementary Information to the main manuscript. These points refer to the third and fourth paragraphs of the Discussion, under 'hypsometric controls on icefield behaviour'.

Point 2 in the first round of reviews questioned whether the dramatic reduction in icefield accumulation area referred to our data or to predicted changes. This is because in the first submission of the manuscript we referred back to another paper by our team that looks at the future mass balance of the icefield (Ing et al., The Cryosphere). In the second submission of the manuscript, for clarity about what our data explicitly and after receiving the reviewers' comments, the reference to Ing et al has been removed.

Therefore, in Point 2, where we are talking about the mass balance changes on the icefield, we are now more clearly only discussing our own data, rather than the predicted changes from Ing et al. The second paragraph under 'Hypsometric controls' now more clearly refers back to the published mass balance work from McNeil et al., 2020 and uses this to contextualise and understand the processes acting on the icefield.

The Reviewer 2 point 3 comment is related, and discusses again the need to clarify if this is referring to our own data or to the mass balance modelling work. We emphasise that we are referring to our own data by linking back to the figure that shows the 2019 AD snowline.

Reviewer #2 (Remarks to the Author):

This is my second time reviewing this article and I am happy that all my suggestions for improvement have been incorporated. As I stated previously, this is a significant piece of work, based on extensive, thoroughly analysed datasets, which support the conclusion and claims. The rearrangement and rewording of the paper has highlighted the significant key findings.

We thank the reviewer for their generosity and time in reviewing the manuscript and for their constructive comments.

A couple of typos:

Line 110: should be "in the same time period" or "at the same time". Amended as requested.

Line 137/8: a 95% confidence interval, rather than the 95%...? Amended as requested.

Line 226: "... the icefield is 315." rather than "of". Amended as requested.

Reviewer #3 (Remarks to the Author):

I would like to thank the authors for having done a great job revising the manuscript following (or, where appropriate, arguing against) the reviewers' comments, and I am very happy with the overall quality of the manuscript! Now even more than before I am convinced that these results should be published, and I see less issues with how some of the results are presented and discussed. The manuscript also reads very well and the figures have been improved substantially, so that I have hardly any minor suggestions. But I still see two points that I would strongly suggest to be changed before publication.

We thank the reviewer for their generous and constructive comments and are delighted that they see the high quality of the manuscript.

The first concerns the claim of a tipping point being reached. The manuscript makes now a much better case arguing that the tipping point is in reach (i.e., that ELAs are approaching the plateau elevations, and are crossing them in individual years), but some of the statements still are too bold. Mostly, I'm concerned with the last sentence of the abstract ("This icefield has now reached a tipping point controlled by a number of feedbacks, with hysteresis inhibiting glacier regrowth."). This is a categorical statement, and while I would agree that it seems very likely that this threshold will be crossed (and in fact might be, and interannual variability clearly already pushes the ELA above threshold elevations), I don't think the claim that the hysteresis is already affecting the SMB is acting right now, is substantiated as clearly in the manuscript as the abstract makes believe. In fact, within the main text, the statements are a lot more careful ("...a decrease in height of a flatter icefield or ice cap will theoretically result in a decrease in surface mass balance (SMB) as the ice surface lowers into warmer air, which reinforces itself as an SMB-elevation positive feedback", or: "the mass balance-elevation feedback will inhibit glacier regrowth (cf.15), likely forming an irreversible tipping point in glacier dynamics"). The melt-elevation feedback is potentially relatively slow. If I compare the rate at which the ELA is rising (from Fig 6i I guess ~200 m over the past 20 years?) to the thinning rate of the plateau (hard to guess from Fig. 7, but from Fig. 6c it should be less than 1 m/yr in recent years), this would indicate that the acceleration in mass loss is clearly dominated by the rise of the ELA (this agrees well with the sentence starting in L356), not by the surface elevation-melt feedback (it might be informative to calculate these rates and compare in the manuscript). I come away with the impression that the authors are looking for a dramatic statement for the abstract and are overshooting – the results they present in the main body of the manuscript are dramatic enough. I simply suggest somewhat more careful wording in the abstract.

We have removed the last sentence of the abstract, and replaced it with a more carefully worded sentence like the one highlighted by the reviewer as appropriate:

“Rising equilibrium line altitudes and increasing ablation across the plateau has driven a series of hypsometrically controlled melt-accelerating feedbacks and resulted in the observed significant acceleration in mass loss. As glacier thinning on the plateau continues, a mass balance-elevation feedback is likely to inhibit future glacier regrowth, potentially pushing glaciers beyond a dynamic tipping point.

The second is still concerning the edited photograph (now Fig. 8). In the end, this is an editorial decision, but I would really strongly advise against publishing an edited photo this way, which to superficial readers (and let's be honest, there are more than enough) may be mistaken as a real one. Unfortunately, in climate science, we additionally have to consider purposeful misunderstandings, and I would detest seeing this figure pop up in social media with the claim that if glaciologists don't find evidence in the field, they resort to editing photos.

The relevant points of the figure could all be made in a schematic drawing of the landscape, and I really don't see any advantage of using an edited photo over a drawn illustration for explaining the concepts of the relevant processes.

Thank you for this suggestion. We are in agreement that it is important to avoid any potential misunderstandings. As suggested by the reviewer, we have replaced the photographs with a schematic drawing of the landscape, highlighting the key processes (see Figure 8). We believe that this will be a useful contribution to the literature; there are very few images that illustrate the conceptualised processes in this way.

One minor point: Please check the legend of Fig 3, there is a double entry (sandur) and I'm not sure all the features in the legend actually occur in the maps (I don't see disconnections and separations).

Thanks for spotting the error in the legend. Legend updated. Figure 3 is also updated to ensure that the disconnections and separations are visible (this was due to incorrect layering of the files in the GIS table of contents).

References

- 1 Johnson, E. & Rupper, S. An Examination of Physical Processes That Trigger the Albedo-Feedback on Glacier Surfaces and Implications for Regional Glacier Mass Balance Across High Mountain Asia. *Frontiers in Earth Science* **8** (2020).
- 2 Marshall, S. J. Regime Shifts in Glacier and Ice Sheet Response to Climate Change: Examples From the Northern Hemisphere. *Frontiers in Climate* **3** (2021). <https://doi.org:10.3389/fclim.2021.702585>
- 3 Nagorski, S. A., Kaspari, S. D., Hood, E., Fellman, J. B. & Skiles, S. M. Radiative Forcing by Dust and Black Carbon on the Juneau Icefield, Alaska. *Journal of Geophysical Research: Atmospheres* **124**, 3943-3959 (2019). <https://doi.org:10.1029/2018JD029411>

- 4 Skiles, S. M., Flanner, M., Cook, J. M., Dumont, M. & Painter, T. H. Radiative forcing by light-absorbing particles in snow. *Nature Climate Change* **8**, 964-971 (2018). <https://doi.org/10.1038/s41558-018-0296-5>
- 5 Kehrwald, N. M. *et al.* Boreal blazes: biomass burning and vegetation types archived in the Juneau Icefield. *Environmental Research Letters* **15**, 085005 (2020).
- 6 Bolibar, J., Rabatel, A., Gouttevin, I., Zekollari, H. & Galiez, C. Nonlinear sensitivity of glacier mass balance to future climate change unveiled by deep learning. *Nature Communications* **13**, 409 (2022). <https://doi.org/10.1038/s41467-022-28033-0>
- 7 Zekollari, H., Huybrechts, P., Noël, B., van de Berg, W. J. & van den Broeke, M. R. Sensitivity, stability and future evolution of the world's northernmost ice cap, Hans Tausen Iskappe (Greenland). *The Cryosphere* **11**, 805-825 (2017). <https://doi.org/10.5194/tc-11-805-2017>
- 8 Schäfer, M., Möller, M., Zwinger, T. & Moore, J. C. Dynamic modelling of future glacier changes: mass-balance/elevation feedback in projections for the Vestfonna ice cap, Nordaustlandet, Svalbard. *Journal of Glaciology* **61**, 1121-1136 (2015). <https://doi.org/10.3189/2015JoG14J184>
- 9 Sass, L. C., Loso, M. G., Geck, J., Thoms, E. E. & McGrath, D. Geometry, mass balance and thinning at Eklutna Glacier, Alaska: an altitude-mass-balance feedback with implications for water resources. *Journal of Glaciology* **63**, 343-354 (2017). <https://doi.org/10.1017/jog.2016.146>
- 10 Millan, R., Mouginot, J., Rabatel, A. & Morlighem, M. Ice velocity and thickness of the world's glaciers. *Nature Geoscience* (2022). <https://doi.org/10.1038/s41561-021-00885-z>
- 11 Ziemen, F. A. *et al.* Modeling the evolution of the Juneau Icefield between 1971 and 2100 using the Parallel Ice Sheet Model (PISM). *Journal of Glaciology* **62**, 199-214 (2016). <https://doi.org/10.1017/jog.2016.13>
- 12 Benn, D. I. & Nicholson, L. Calculating ice melt beneath a debris layer using meteorological data. *Journal of Glaciology* **52**, 463-470 (2006). <https://doi.org/10.3189/172756506781828584>
- 13 Davies, B. *et al.* Topographic controls on ice flow and recession for Juneau Icefield (Alaska/British Columbia). *Earth Surface Processes and Landforms* **47**, 2357-2390 (2022).
- 14 Pelto, M. Exceptionally High 2018 Equilibrium Line Altitude on Taku Glacier, Alaska. *Remote Sensing* **11**, 2378-2378 (2019).
- 15 U.S. Geological Survey, B. G. P. (2016).
- 16 McNeil, C. *et al.* Explaining mass balance and retreat dichotomies at Taku and Lemon Creek Glaciers, Alaska. *Journal of Glaciology* **66**, 530-542 (2020). <https://doi.org/10.1017/jog.2020.22>
- 17 O'Neel, S. *et al.* Reanalysis of the US Geological Survey Benchmark Glaciers: long-term insight into climate forcing of glacier mass balance. *Journal of Glaciology* **65**, 850-866 (2019). <https://doi.org/10.1017/jog.2019.66>

Reviewer #1 (Remarks to the Author):

After reading the response to referees letter and the updated version of the manuscript, I would like to congratulate manuscript authors as I consider now the work ready for publication. The results have high interest for the scientific community and are conveniently supported by the data and the analysis. Moreover, some changes (i.e. new figure 8, albedo trend analysis in the supplementary material...) are highly suitable to show the main conclusions. Moreover, I can say that I have enjoyed the read of the final manuscript. Thanks for your effort and your time on rising this article.

Reviewer #3 (Remarks to the Author):

I would like to thank the authors for their effort to better express the somewhat hypothetical nature of their findings regarding the importance of the different feedbacks for the acceleration of mass loss. Their responses and edits to the manuscript (including the new design of Fig. 8) are adequate and I now find the manuscript of suitable quality to be published. I have no remaining questions or comments.